# Voluntary control of semantic neural representations by imagery with conflicting visual stimulation

Ryohei Fukuma [1,2,3], Takufumi Yanagisawa [1,2,3,4 ✉], Shinji Nishimoto [5,6], Hidenori Sugano [7], Kentaro Tamura[8], Shota Yamamoto [1], Yasushi Iimura[7], Yuya Fujita [1], Satoru Oshino [1], Naoki Tani [1], Naoko Koide–Majima[5,6], Yukiyasu Kamitani[2,9] & Haruhiko Kishima [1,4]

Neural representations of visual perception are affected by mental imagery and attention. Although attention is known to modulate neural representations, it is unknown how imagery changes neural representations when imagined and perceived images semantically conflict. We hypothesized that imagining an image would activate a neural representation during its perception even while watching a conflicting image. To test this hypothesis, we developed a closed-loop system to show images inferred from electrocorticograms using a visual semantic space. The successful control of the feedback images demonstrated that the semantic vector inferred from electrocorticograms became closer to the vector of the imagined category, even while watching images from different categories. Moreover, modulation of the inferred vectors by mental imagery depended asymmetrically on the perceived and imagined categories. Shared neural representation between mental imagery and perception was still activated by the imagery under semantically conflicting perceptions depending on the semantic category.

[1] Department of Neurosurgery, Graduate School of Medicine, Osaka University, Suita, Japan. [2] ATR Computational Neuroscience Laboratories, Seika-cho, Japan. [3] Institute for Advanced Co-Creation Studies, Osaka University, Suita, Japan. [4] Osaka University Hospital Epilepsy Center, Suita, Japan. [5] Center for Information and Neural Networks (CiNet), National Institute of Information and Communications Technology (NICT), Suita, Japan. [6] Graduate School of Frontier Biosciences, Osaka University, Suita, Japan. [7] Department of Neurosurgery, Juntendo University, Tokyo, Japan. [8] Department of Neurosurgery, Nara Medical University, Kashihara, Japan. [9] Graduate School of Informatics, Kyoto University, Kyoto, Japan. ✉email: tyanagisawa@nsurg.med.osaka-u.ac.jp

Neural activities in the visual cortex reflect both externally driven bottom-up sensory information and internally generated top-down signals, such as mental imagery[1,2] and attention[3]. Neural decoding using machine-learning techniques has revealed relations between top-down signals and bottom-up information by evaluating their neural representations in the visual cortex. Perceived images can be inferred (decoded) from visual cortical activities as reconstructed images[4–6] or semantic attributes of the images[7,8]. A trained decoder for the perceived images can successfully infer mental imagery content[5,9], establishing that perception and imagery have common neural representations in the early[9] and higher[5,10,11] visual cortices. Similarly, attention can be inferred by the decoder for perceived images[12]. These results suggest that there are common neural representations for both externally driven bottom-up sensory information and internally generated top-down signals of mental imagery and attention.

However, little is known about how the bottom-up information and the top-down signals related to mental imagery interact with each other with regard to neural representations. Previous studies revealed that attention modulates neural representations[3] on the basis of semantic similarity to the category of the attended object[13]. In addition, behavioral studies have suggested that imagery acts like a weak perception in facilitating subsequent perception[14,15]. However, it is not known how the neural representations of perceiving images are modulated by imagery when they semantically conflict with each other. For example, when people imagine a human face while watching a landscape such as a mountain view, it remains unclear how the resulting neural representations differ from the neural representations while watching the same landscape without imagining something (e.g., a human face). Here, we hypothesized that imagining an image would activate a neural representation of perceiving the imagined image even while watching a conflicting image. We evaluated how neural representations resulting from watching an image in category (A) become closer to those in a different category (B) by imagining an image in category (B).

To compare the neural representations while perceiving and imagining various categories of images, we applied neural decoding to electrocorticograms (ECoGs). ECoGs are characterized by high temporal resolution and wide coverage of the cortices[16]; these characteristics make ECoGs suitable for evaluating visual information that is sparsely represented in the visual cortex[7]. Moreover, ECoGs have been used to infer several semantic categories of visual stimuli[17–19]. In this study, neural decoding was combined with a visual semantic space in which the semantic attributes of an image were embedded into a vector representation[7]; therefore, the changes in neural representations were evaluated as changes in semantic vectors.

Moreover, to explore the effects of imagery on neural representations while perceiving and imagining different images from various categories, we developed a closed-loop system in which the subject was presented images that corresponded to semantic vectors inferred from real-time neural activities. The semantic vector was inferred by a decoder trained on neural activities when the subject watched images from various semantic categories. Accordingly, if a subject views a feedback image of a certain category without any imagery or attention, images of the same category will be displayed in the closed-loop system. In contrast, if our hypothesis is true for a certain category, imagining an image from that category would make the semantic vector inferred from the neural activities closer to the semantic vector of the imagined category while watching images from different categories, resulting in the display of an image closer to the imagined category. Even if the change in the semantic vector due to the imagery is small, successive changes in the semantic vectors

in the same direction will eventually show the image representing the imagined category. We refer to this intentional control of the semantic vector represented by a feedback image as representational brain-computer interaction (rBCI), in which the feedback images are embedded in a representational space and controlled through the interaction between the decoding-based visual feedback and the top-down intention to alter the feedback. Using the rBCI, we evaluated how neural representations are modulated by imagery in the presence of various types of conflicting bottom-up sensory information.

In this study, we demonstrated that subjects can control feedback images from an rBCI using ECoGs of the visual cortical areas by intending to show images related to specific semantic categories. Moreover, for categories in which the subjects successfully controlled the feedback image, we also demonstrated that the ECoGs during perception of a particular category are modulated by the imagery associated with a different category, resulting in a semantic vector inferred from the ECoGs that was closer to the imagined category.

## Results

### ECoG recordings and experimental procedure.
ECoGs were recorded from 21 subjects with epilepsy (E01–E21) who had subdural electrodes implanted on their occipital or temporal lobes, including the ventral visual cortex (Fig. 1a and Supplementary Figs. 1 and 2a; also see Supplementary Table 1). Among them, 17 subjects (E01–E17) watched six 10-min videos (training videos) consisting of short movies with various semantic attributes (video-watching task). In addition, 12 of these subjects (E01, E03, E06, E07, and E09–E16) watched a 10-min video (validation video) consisting of four repetitions of a 2.5-min movie that was different from the movies in the training video. Based on ECoGs obtained while subjects watched the training video, we constructed a decoder to infer the semantic attributes of the presented scenes. Then, for four subjects (E01–E04), we applied the decoder in a closed-loop condition to present images that were selected based on the semantic attributes inferred from the ECoGs (Fig. 1b). Last, for 13 subjects (E01–E09 and E18–E21), we recorded ECoGs while the subjects watched images from a particular category with and without imagery associated with a different category.

### High-γ features of ECoGs respond to semantic attributes of movies.
Initially, the ECoG frequency bands that consistently responded to the videos were evaluated by the replicability of their power while subjects watched the repeated movies in the validation video. Power in four frequency bands (α, 8–13 Hz; β, 13–30 Hz; low γ, 30–80 Hz; high γ, 80–150 Hz) was calculated for nonoverlapping 1000-ms time windows from the ECoGs during the validation video and were compared across the repeated movies by Pearson's correlation coefficients. Among the four frequency bands, power in the high-γ band responded most consistently to the video stimuli in the early visual area (V1–V4) and the higher visual area (middle temporal complex and neighboring visual areas, ventral stream visual cortex, medial temporal cortex, and lateral temporal cortex) (Fig. 2a; for consistency depending on the time window, see Supplementary Fig. 2b).

Semantic attributes of each scene in the training videos were embedded into a 1000-dimensional visual semantic space to reveal how the normalized high-γ powers (features) responded to the semantic attributes of the videos. First, the training videos were converted into still images at one-second intervals (3600 images) and annotated by cloud workers. Then, the annotation of each scene was converted into a 1000-dimensional vector in the

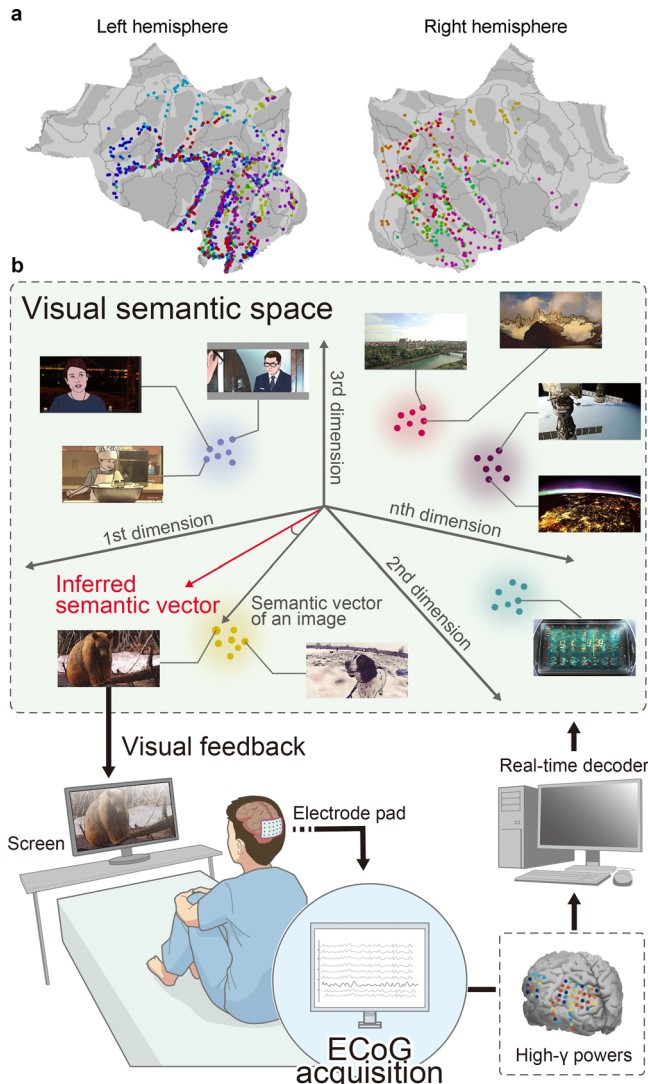

**Fig. 1 Location of electrodes and schematic of a closed-loop system.**
**a** Locations of subdural electrodes are color-coded for each subject who participated in this study ($n = 21$). **b** ECoGs were acquired in real time to infer a semantic vector of 1000 dimensions in the visual semantic space from power in the high-γ band. An image that had the nearest semantic vector to the inferred semantic vector was presented to the subject as a feedback image. Because of copyright restrictions, the actual images used in the tasks have been replaced with illustrations throughout this paper. Details of the creation of the illustrations are presented in "Methods".

space learned by a word-embedding model called the skip-gram model[20] ($V_{true} := \{v^i_{true} | i = 1, \cdots, 3600\,(\text{scene})\}$; see "Methods" and Supplementary Fig. 2c; also see Supplementary Fig. 2d for the consistency of the semantic vectors across annotators). When we applied principal component analysis (PCA) to the proposed semantic vectors of the training videos ($V_{true}$), the first and second principal components contrasted "human face" scenes from other scenes and "landscape" scenes from "word" scenes, respectively (Fig. 2b, Supplementary Fig. 2e, f, and Supplementary Table 2). To clarify the distribution of scenes in the first and second principal components, we selected 50 scenes with the highest Pearson's correlation coefficient among the 1000 values of $v^i_{true}$ and $v_{category}$ ($R(v_{category}, v^i_{true})$) ($v_{category}$: $v_{word}$, $v_{landscape}$, or $v_{face}$; the semantic vector of the word "word", "landscape", or an average of the semantic vectors of "human" and "face") from the 3600 scenes of the training video. The selected 50 scenes in each

category were separately distributed in the first and second principal components of the semantic vectors (Fig. 2b; also see Supplementary Fig. 2g). $V_{true}$ successfully captured these semantic attributes of the presented videos.

The high-γ features during the presentation of the selected scenes in the training videos were compared to elucidate cortical regions that differentially responded to the categories. The high-γ features were calculated from the 1000-ms ECoGs centered at the time when the annotated image of the selected scenes was presented. The high-γ features in the higher and early visual areas significantly differed depending on the categories ($P < 0.05$, $n = 50$ for each group, one-way analysis of variance [ANOVA], adjusted using the Benjamini–Hochberg procedure[21]; Fig. 2c; for partial $\eta^2$, see Supplementary Fig. 2h). The high-γ features in the visual areas differentially responded to the semantic categories.

To evaluate how the high-γ features responded to the semantic attributes while excluding the contribution of these low-level features such as contrast and sounds, we inferred the high-γ features from each electrode based on the semantic features (semantic vectors) and low-level visual and auditory features from the training videos using ridge regression with a tenfold nested cross-validation. By applying motion energy filters[8,22] to the videos and modulation-transfer function models[23] to the sound of the videos, 2139 low-level visual features, and 2000 low-level auditory features were acquired for each scene in the training video (see "Methods"). The high-γ features from the visual areas were significantly explained by semantic features and low-level visual and auditory features (Fig. 2d). Even when the regression weights for the low-level visual and auditory features were set to zero, the high-γ features in the visual areas were still significantly explained by the semantic features ($P < 0.05$, $n = 3600$ for each electrode, one-sided Pearson's correlation test, adjusted using the Benjamini–Hochberg procedure for each feature; Fig. 2d), suggesting that the high-γ features in these visual areas responded not only to the low-level visual and auditory features of the training videos but also to the semantic attributes of the videos that were represented by the semantic vectors.

**Decoding of semantic vectors corresponding to the presented scenes.** The semantic vector for the $i$th scene ($v^i_{true}$) in the training videos was inferred from the high-γ features using ridge regression with tenfold nested cross-validation. The scenes for training and testing in the cross-validation were selected from different movie sources (see "Methods" and Supplementary Fig. 3a). The accuracy of inferring the semantic vector was evaluated for each principal component of $V_{true}$. The inferred semantic vectors ($V_{inferred} := \{v^i_{inferred} | i = 1, \cdots, 3600\,(\text{scene})\}$) were projected to the $k$th direction vector of the PCA ($k = 1, \cdots, 1000$) so that the Pearson's correlation coefficient between the projected values and the $k$th principal component of $V_{true}$ were calculated for the 3600 scenes to obtain the projected correlation coefficient ($PrjR^k$ ($V_{inferred}$, $V_{true}$)). For 14 principal components, $PrjR^k$ ($V_{inferred}$, $V_{true}$) showed a significant positive correlation (Fig. 3a and Supplementary Fig. 3b–d). It should be noted that $PrjR^k(V_{inferred}$, $V_{true}$) was especially high for the first several principal components.

In addition, for the previously selected 50 scenes from among 3600 scenes in the training videos for each of the three categories, we evaluated the accuracy of classifying the category of the presented scene from the inferred semantic vector. For each semantic vector inferred from the high-γ features from all implanted electrodes ($v^i_{inferred}$), Pearson's correlation coefficients with the semantic vectors of the categories were calculated ($R(v_{category}, v^i_{inferred})$), where $v_{category}$ was $v_{word}$, $v_{landscape}$, or $v_{face}$).

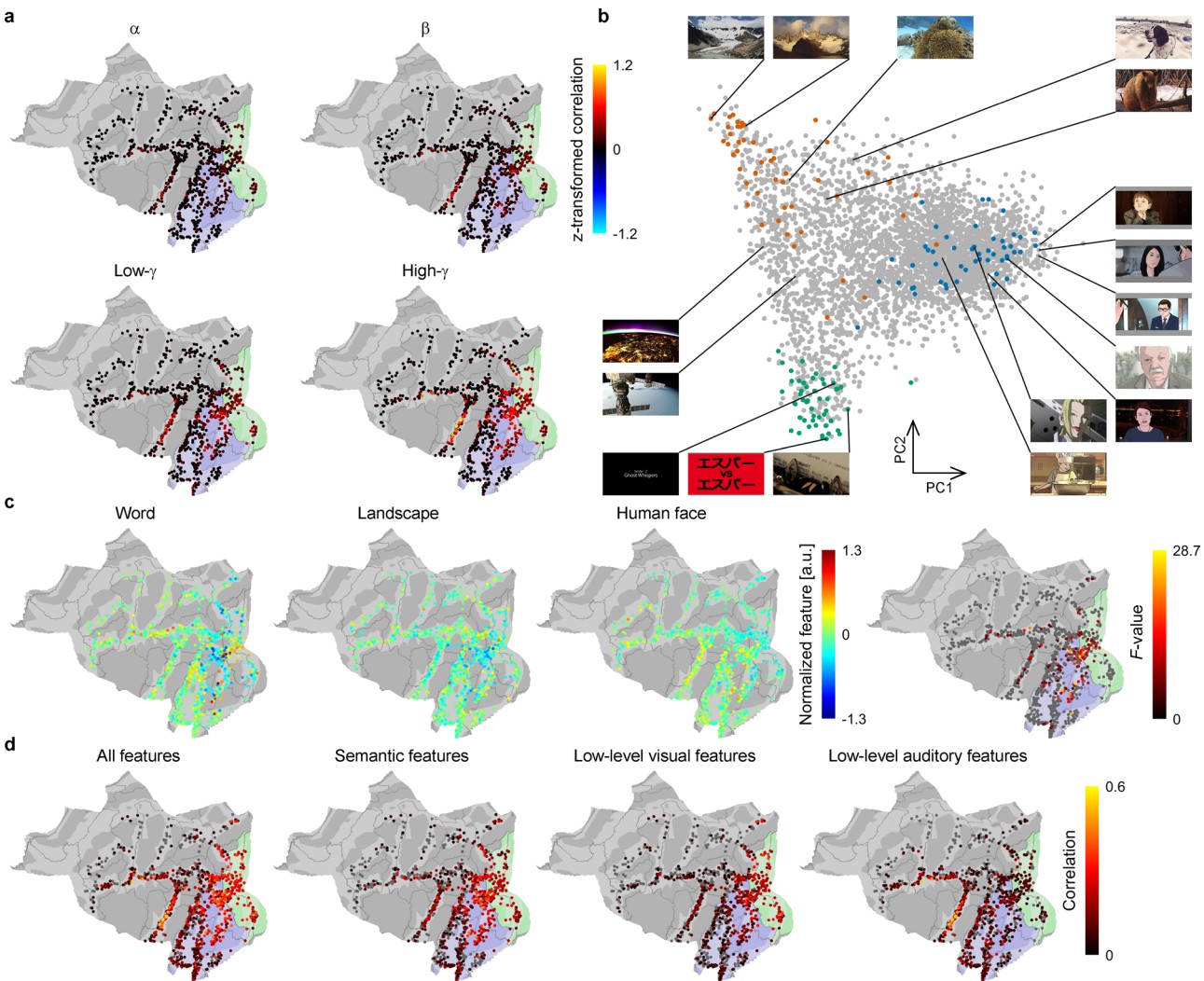

**Fig. 2 Visual semantic space and high-γ powers/features in the video-watching task. a** The Fisher z-transformed Pearson's correlation coefficients between the 150 powers corresponding to repeated 2.5-min movies in the validation video were averaged across all possible combinations of the repetitions to be color-coded on each electrode. Areas shaded with blue and green denote the higher and early visual areas, respectively. **b** Each scene in the training videos is shown at a position based on the first and second principal components (PCs) of the semantic vectors. Green, red, and blue points represent the positions of 50 scenes selected as representative of the categories of "word", "landscape", and "human face", respectively. **c** In the training videos, the high-γ features corresponding to the 50 selected scenes from the three categories were averaged within each category and were color-coded at the location of the electrodes. For visibility, features of each electrode were z-scored within 3600 scenes. The F-values of ANOVA for the high-γ features were similarly color-coded at each electrode ($P < 0.05$, $n = 50$ for each group, one-way ANOVA, adjusted using the Benjamini–Hochberg procedure). **d** Pearson's correlation coefficients between the high-γ features while subjects watched the training videos and the inferred high-γ features were color-coded and shown at the location of the electrode. For each outer fold of the tenfold nested cross-validation, a decoder (ridge regression model) was first trained to infer high-γ features using all features (semantic features, and the low-level visual and auditory features) from the videos; then, for each feature set, the weights corresponding to the other two feature sets were set to zero before the regression model was applied to the test data. A correlation map of each feature set was calculated from the entire 3600 scenes. Only electrodes that showed significant positive correlations are shown ($P < 0.05$, $n = 3600$ for each electrode, one-sided Pearson's correlation test, adjusted using the Benjamini–Hochberg procedure for each feature).

The classification was considered to be correct when the $R(v_{category}, v^i_{inferred})$ was the highest for the category of the presented scene. The accuracies to classify two of the three categories (binary accuracies) were $67.5 \pm 4.7\%$ (mean $\pm$ 95% confidence intervals [CIs] among subjects) for word versus landscape, $70.8 \pm 4.8\%$ for landscape versus human face, and $73.1 \pm 4.5\%$ for human face versus word, all of which were significantly higher than chance level (50%) ($P < 1.0 \times 10^{-6}$, $n = 17$, one-sided one-sample $t$-test with Bonferroni-adjusted α-level of 0.0167 [0.05/3]; for accuracy in the higher and early visual areas, see Fig. 3b). The accuracy across the three categories was $56.2 \pm 5.6\%$ (also see Supplementary Fig. 3e–g). The three

categories of the presented scenes were successfully classified by the high-γ features.

**Control of inferred images in the closed-loop condition.** Four subjects (E01–E04) participated in a real-time feedback task to control the rBCI using the online decoder that was trained with the high-γ features for all 3600 scenes of the training videos (Fig. 1b and Fig. 4a, b; also see Supplementary Fig. 4a–d for the accuracy and consistency in the validation video using the online decoder). Prior to this task, the subjects were first informed that images would be presented based on their real-time brain activity, and they were instructed to display a feedback image representing

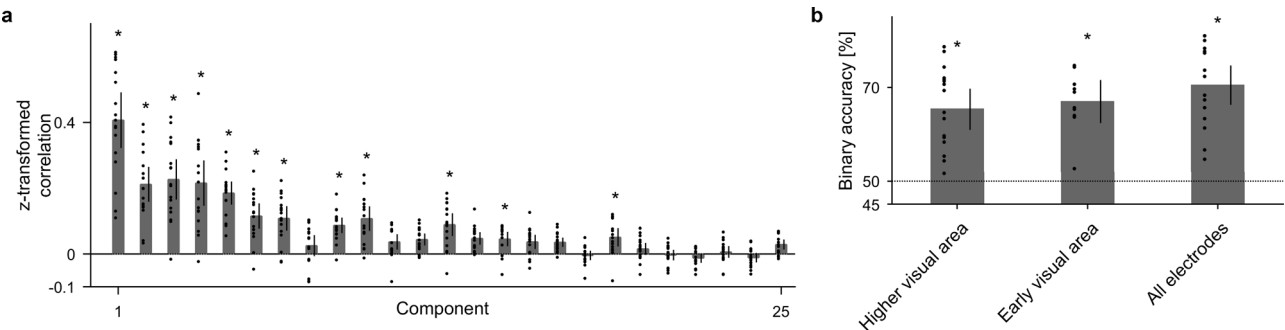

**Fig. 3 Decoding accuracy in the video-watching task. a** $PrjR^k(V_{inferred}, V_{true})$ was Fisher z-transformed and averaged across the 17 subjects ($z(PrjR^k(V_{inferred}, V_{true}))$), which is shown in the order of principal components. For visibility, the first 25 components are shown (for all components, see Supplementary Fig. 3b). Individual values are shown with dots. Error bars denote 95% CIs among the subjects. *$P < 0.5 \times 10^{-4}$ (Bonferroni-adjusted α-level; 0.05/1000), two-sided permutation test. **b** Binary classification accuracies for all three pairs from the three categories (word, landscape, and human face) were averaged to show the subject-averaged binary classification accuracy with the bars. Individual values are shown with dots. Error bars denote 95% CIs among subjects. The accuracy was calculated based on the semantic vectors inferred from high-γ features from the higher visual area ($n = 17$), early visual area ($n = 10$), and all implanted electrodes ($n = 17$). There was no significant difference between the accuracies based on the higher and early visual areas ($P = 0.5767$, $t(23.3) = -0.57$, uncorrected two-sided Welch's $t$-test). *$P < 1.7 \times 10^{-2}$ (Bonferroni-adjusted α-level; 0.05/3), one-sided one-sample $t$-test against chance level (50%).

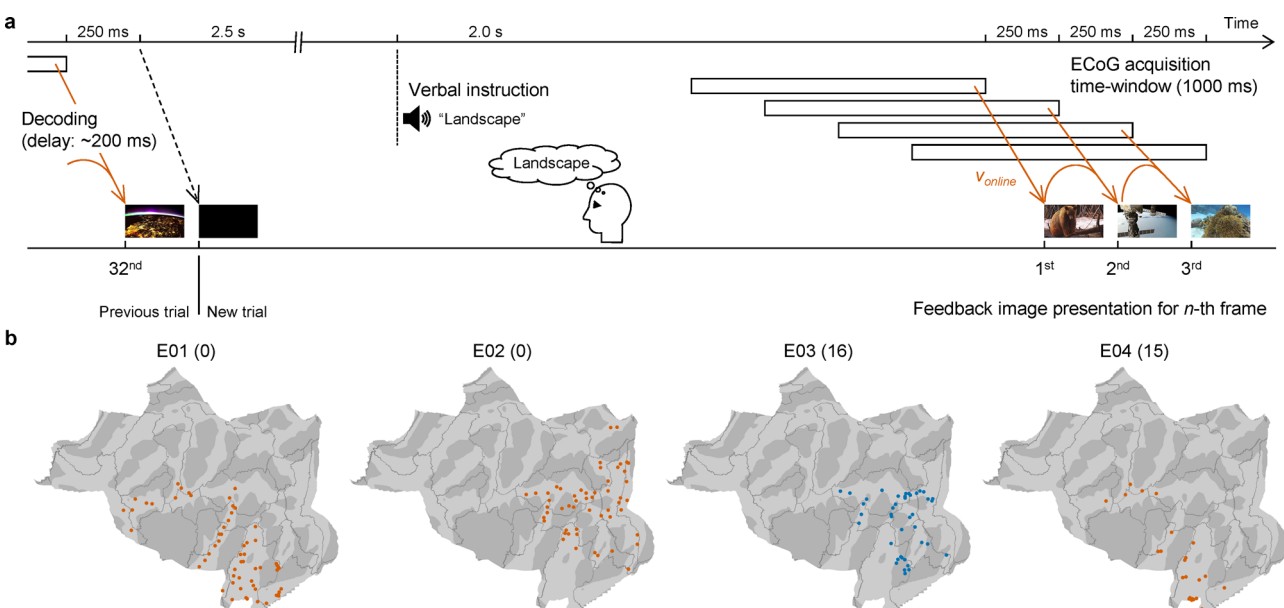

**Fig. 4 Real-time feedback task. a** Timing of ECoG acquisition and feedback image presentation during the real-time feedback task is shown with a schematic. Immediately after delivery of the instruction, 1000-ms ECoGs were acquired while the subject was watching the black screen to infer the semantic vector in real-time (online vector: $v_{online}$) with the online decoder. Next, the first feedback image was selected from among the images in the training videos based on the highest $R(v_{online}, v^i_{true})$ and presented to the subject. Then, successive 1000-ms ECoGs were acquired every 250 ms to calculate the next online vector ($v_{online}$), which was the linear interpolation between the inferred vector from the ECoGs and the previous online vector. A total of 32 feedback images were presented for each instruction. The order of instructions was randomized. The system delay from the end of the acquisition to the image presentation was approximately 200 ms. All four subjects participated in four sessions of 30 trials each, with breaks between sessions to minimize their fatigue. **b** Locations of subdural electrodes used for decoding in the real-time feedback task are mapped on a normalized brain surface. Red and blue markers denote electrodes in the left and right hemispheres, respectively. The number of depth electrodes used in the real-time decoding is shown in parentheses.

a particular category (i.e., word, landscape, or human face) by visually imagining it. In particular, they were asked to maintain displaying images related to the particular category as long as possible. It is worth noting that subjects were instructed to freely imagine images that they felt well represented the particular category; that is, we did not specify images to be imagined. At the beginning of each trial, a black screen was displayed, and one of the categories (target category) was given orally in Japanese (Fig. 4a). High-γ features of 1000-ms ECoGs were calculated to

infer the semantic vector in real-time (online vector: $v_{online}$) with the online decoder. The feedback image was selected based on the highest $R(v_{online}, v^i_{true})$ in the 1000-dimensional semantic space.

Figure 5a shows a representative result for three consecutive trials in the real-time feedback task (cf. Supplementary Movies 1–4). The accuracy of the online control was evaluated as a three-choice task based on the Fisher z-transformed correlation coefficient between $v_{online}$ and $v_{target}$ ($z(R(v_{online}, v_{target}))$; $v_{target}$: semantic vector of the target category, $v_{word}$, $v_{landscape}$, or $v_{face}$). When we defined a

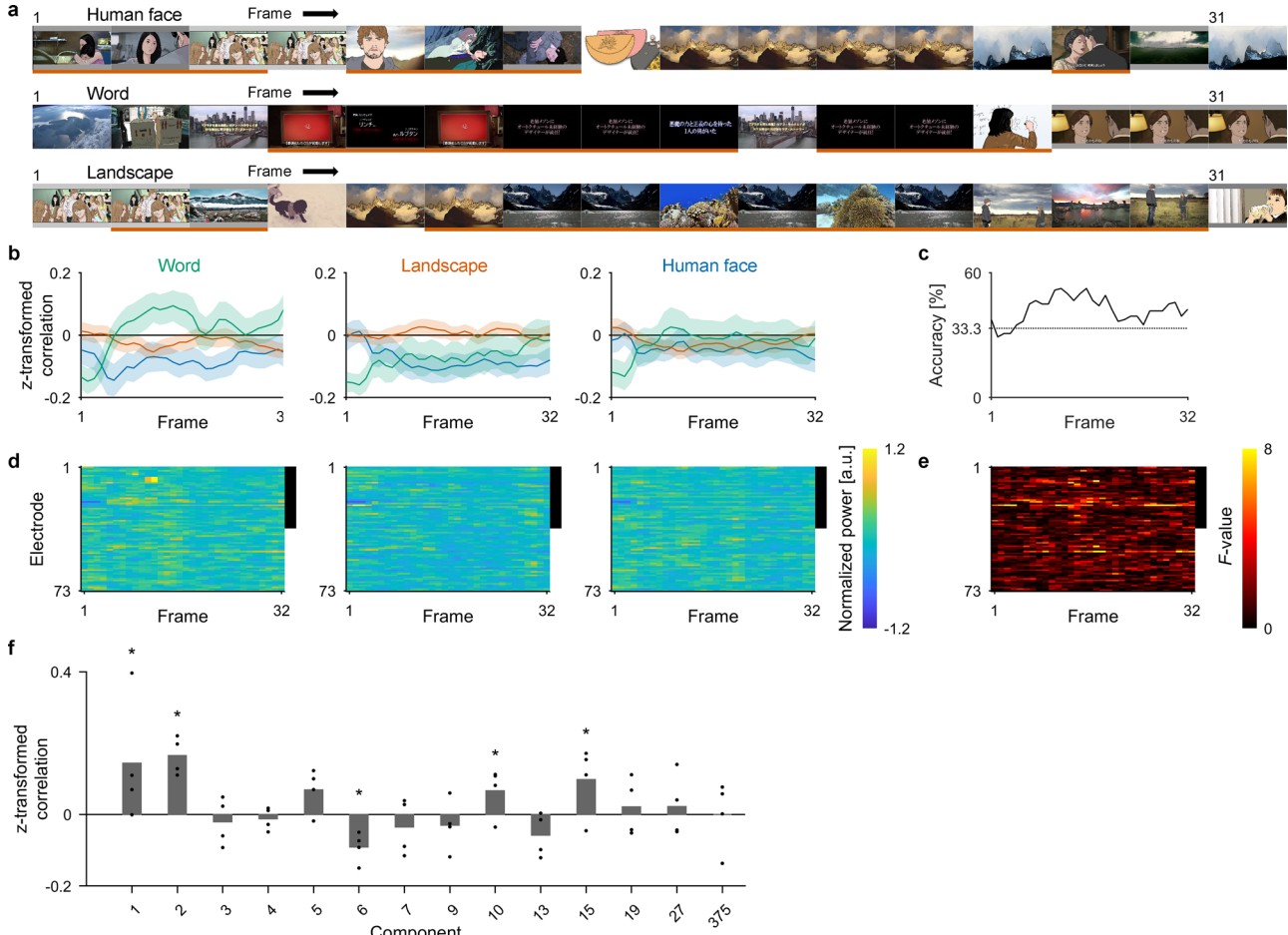

**Fig. 5 Closed-loop performance in controlling the inferred images. a** Representative feedback images are shown for E01 during trials 28–30 of the first session with the corresponding target categories shown on the top. Due to the figure size, one of every two images is shown. The images underlined in red are the correct decoding in the context of a three-choice task in each frame; because the evaluation of the three-choice task is based on the inferred vector ($v_{online}$), some frames have the same feedback images, but are differently classified. **b** For E01, the trial-averaged $z(R(v_{online}, v_{category}))$, where ($v_{category}$: $v_{word}$, $v_{landscape}$, or $v_{face}$), are shown as green, red and blue lines, respectively, with 95% CIs by the shaded area. The title of each panel indicates the target categories for the averaged trials. **c** The three-choice accuracy at each frame is shown. Dotted line denotes chance level (33.3%). **d** Power in the high-γ band during the feedback trials for each target category were averaged to be color-coded for each electrode and frame. The black line on the right side of the plot indicates the electrodes in the higher visual area. For visualization, the powers were z-scored across all trials and frames for each electrode. **e** For each electrode and frame, $F$-values of one-way ANOVA across the high-γ powers during the three target categories in (**d**) were color-coded. The black line on the right side of the plot indicates the electrodes in the higher visual area. **f** $PrjR^k$ ($V_{online}$, $V_{target}$) ($V_{online} := \{v_{online}^{i,j}\}$ and $V_{target} := \{v_{target}^{i,j}\}$ where $i = 1, \cdots, 120$ (trial); $j = 1, \cdots, 32$ (frame)) were Fisher z-transformed and averaged across all four subjects to be shown in order of the principal components. Here, $PrjR^k$ ($V_{online}$, $V_{target}$) was evaluated only for the 14 principal components whose $z(PrjR^k(V_{inferred}, V_{true}))$ was positively significant in the video-watching task. The $z(PrjR^k(V_{online}, V_{target}))$ was compared with the corresponding chance distribution for each component ($k$) ($z(PrjR^k(V_{online}, \{v_{target}^{i,j}\}))$ where $i$ is shuffled (1, $\cdots$, 120); $j = 1, \cdots, 32$). Individual values are shown with dots. *$P < 3.6 \times 10^{-3}$ (Bonferroni-adjusted α-level; 0.05/14), two-sided permutation test.

successful trial as that in which frame-averaged $\overline{z(R(v_{online}, v_{target}))}$ was larger than the other two $\overline{z(R(v_{online}, v_{nontarget}))}$ ($v_{nontarget} \in \{v_{word}, v_{landscape}, v_{face}\} \backslash \{v_{target}\}$), the trials with target categories of word and landscape in Fig. 5a were successful. Overall, E01 succeeded in following instructions with a three-choice accuracy of 45.83%, which was significantly greater than chance ($P = 0.0021$, one-sided permutation test; word vs. landscape, 70.00%; landscape vs. human face, 61.25%; human face vs. word, 58.75%). For the other three subjects, the accuracies were also significant (E02: 50.00%, $P < 0.0001$; word vs. landscape, 55.00%; landscape vs. human face, 73.75%; human face vs. word, 72.50%; E03: 41.67%, $P = 0.031$; word vs. landscape, 62.50%; landscape vs. human face, 46.25%; human face vs. word, 58.75%; E04: 41.67%, $P = 0.0065$; word vs. landscape, 60.00%; landscape vs. human face, 52.50%; human face vs. word, 60.00%; for comments from the interviews

after each session, see Supplementary Table 3). Therefore, all four subjects succeeded in controlling the inferred semantic vector to be closer to the instructed category.

During the real-time feedback task, E01 succeeded in increasing $z(R(v_{online}, v_{target}))$ more than other two $z(R(v_{online}, v_{nontarget}))$ under the target category of word and landscape (Fig. 5b), resulting in the three-choice accuracy at each frame increasing up to 52.5% in the middle of the trials (Fig. 5c, Supplementary Fig. 5a–c, and Supplementary Table 4). Along with this increase in accuracy, the high-γ power from electrodes in the higher visual area differed depending on the instructions (Fig. 5d), and these differences were associated with high one-way ANOVA $F$-values (Fig. 5e; also see Supplementary Fig. 5d, e).

The accuracy to control the online vector during the real-time feedback task was also evaluated in the 1000-dimensional

semantic space. The projected correlation coefficients between the online vector and the semantic vector of the target category $(z(PrjR^k(V_{online}, V_{target})))$ were positively significant for four principal components (#1, #2, #10, and #15) ($P < 0.05$, $n = 4$, two-sided permutation test with Bonferroni-adjusted α-level of $3.6 \times 10^{-3}$ [0.05/14]; Fig. 5f). The online vectors were successfully controlled along several dimensions in the semantic space in this closed-loop condition.

**Modulation of inferred vectors by mental imagery**. To evaluate the degree to which the inferred semantic vectors could be modulated by mental imagery, an imagery task was performed by 13 subjects (E01–E09 and E18–E21) to compare the ECoGs while watching an image with or without imagining an image from a different category. From among the 3600 annotated images in the training videos, we selected five images representing "word" and five images representing "landscape" that had high $R(v^i_{true}, v_{word})$ and $R(v^i_{true}, v_{landscape})$, respectively, and did not have content relating to another meaning (Supplementary Fig. 6a). In each trial, the subject was first presented an image of a word or landscape for 2 s to memorize (i.e., nonimagery period) and was then presented an image from a different category for another 2 s (i.e., imagery period); during the imagery period, the subject was instructed to visually imagine the first image while watching the second image (Fig. 6a).

The high-γ powers while watching the images from one category were altered by imagining an image from another category. Figure 6b shows a representative time–frequency map of an electrode implanted at V1 in E05. Powers in the high-γ band increased ~100–1000 ms after the presentation of images representing "word" and "landscape" without any imagery; however, when imagining an image from another category, power in the same high-γ frequency band decreased, although the subjects were watching the same image. Similarly, decreases in power in the high-γ band were observed in the early visual area across the 13 subjects (Fig. 6c). The imagery altered power in the high-γ band although the same images were being watched.

A decoder was trained for each of the nine subjects who participated in both the video-watching task and the imagery task (E01–E09). The decoder was trained using the high-γ features of the subdural electrodes while the subject was watching the training videos; subsequently, the decoder used the high-γ features of the same electrodes to infer a semantic vector ($v_{inferred}$) from 0 to 1 s after the presentation of each image in the imagery task. Figure 7a shows the $R(v_{inferred}, v_{word})$ and $R(v_{inferred}, v_{landscape})$ for each image presented to E01. In the nonimagery period, each category of images was distributed separately so that two categories of images were classified with areas under the curves (AUCs) of 0.9248 and 0.7936 from the receiver operating characteristic curves of $R(v_{inferred}, v_{word})$ and $R(v_{inferred}, v_{landscape})$, respectively (binary accuracy between the two categories: 80.00%); however, in the imagery period, the distribution of the landscape image moved toward the word image. In fact, the $z(R(v_{inferred}, v_{word}))$ while imagining word images and watching a landscape image significantly increased compared to that while watching a landscape image without the imagery ($\Delta Z_{word} = 0.1570$, $P = 0.0006$, $t(47.81) = 3.44$, $n = 25$ for each group, uncorrected one-sided Welch's $t$-test; Supplementary Fig. 6b); in contrast, the $z(R(v_{inferred}, v_{landscape}))$ while imagining landscape images and watching a word image was not significantly increased compared to that while watching a word image without the imagery ($\Delta Z_{landscape} = -0.0040$, $P = 0.57$, $t(47.10) = -0.17$).

Among all nine subjects included in this analysis, the inferred semantic vectors were modulated in the direction of the semantic vectors of "word" and "landscape" by imagery.

Initially, $R(v_{inferred}, v_{word})$ and $R(v_{inferred}, v_{landscape})$ successfully classified the images of words and landscapes during the nonimagery period using the high-γ features of the subdural electrodes from 0 to 2.0 s (Fig. 7b; binary accuracy between the two categories for 0–1.0 s: 80.8 ± 9.0% [mean ± 95% CIs among subjects]; for AUC during the imagery period, see Supplementary Fig. 6c). The $z(R(v_{inferred}, v_{word}))$ with the landscape image significantly increased while imagining word images using the high-γ features from 0 to 1.0 s and 1.0 to 2.0 s ($n = 9$, one-sided one-sample $t$-test with Bonferroni-adjusted α-level of 0.0083 [0.05/6]; Fig. 7c; for the difference in the correlation coefficient for the perceived category attributable to the imagery, see Supplementary Fig. 6d). Similarly, the $z(R(v_{inferred}, v_{landscape}))$ with the word image significantly increased while imagining landscape images using the high-γ features from 0.5 to 1.5 s (Fig. 7c; for other frequency bands, see Supplementary Fig. 6e, f). In addition, although the accuracies to classify the images of "word" and "landscape" showed a tendency to be higher with the early visual area than with the higher visual area (Fig. 7d), the modulations using the high-γ features from the higher visual area showed a tendency to be higher than those from the early visual area (Fig. 7e). Similar to the modulations using the high-γ features of all subdural electrodes (Fig. 7c), $\Delta Z_{word}$ increased immediately after image presentation (0–1.0 s), whereas $\Delta Z_{landscape}$ increased at 1 s after image presentation (0.5–1.5 s) using the high-γ features from the higher visual area. Therefore, it was demonstrated that the modulation of the semantic vector depends on time from the initiation of imagery, the imagined category, and/or the category of the presented image (for the results of the representational similarity analysis, see Supplementary Fig. 7). Moreover, different anatomical areas were suggested to contribute to the modulations.

## Discussion

This study tested the hypothesis that imagining an image from one category while watching a conflicting image from another category results in neural representations closer to those obtained while watching the image of the imagined category. Using rBCI, the subjects succeeded in controlling the inferred semantic vector to move closer to the semantic vector associated with the instructed category. The rBCI enabled an exploratory search for the semantic category for which the hypothesis stood because the successful control of the feedback image required that the semantic vector inferred from the neural activities becomes closer to the semantic vector corresponding to the imagined category while feedback images from the various categories were presented to the subject. Then, for the semantic categories successfully controlled with rBCI, modulation of the semantic vector by imagery was evaluated in the imagery task. The imagery task revealed that the inferred vector was modulated such that it moved significantly closer to the imagined category even when watching an image from a different category, although this modulation depended on the particular image category and time from the initiation of imagery. These results supported our hypothesis, at least regarding the imagery associated with words and landscapes.

Mental imagery[1,2] and attention[3] are possible top-down mechanisms by which neural representations are intentionally modulated[24] to achieve rBCI control of the inferred images. In fact, during the real-time feedback control, the subjects tended to focus their attention on specific parts of the feedback image that were close to the instructed category (e.g., subtitles in the images when the word instruction was given; see Supplementary Table 3), although we instructed them to imagine the instructed category. However, it is difficult to explain feedback

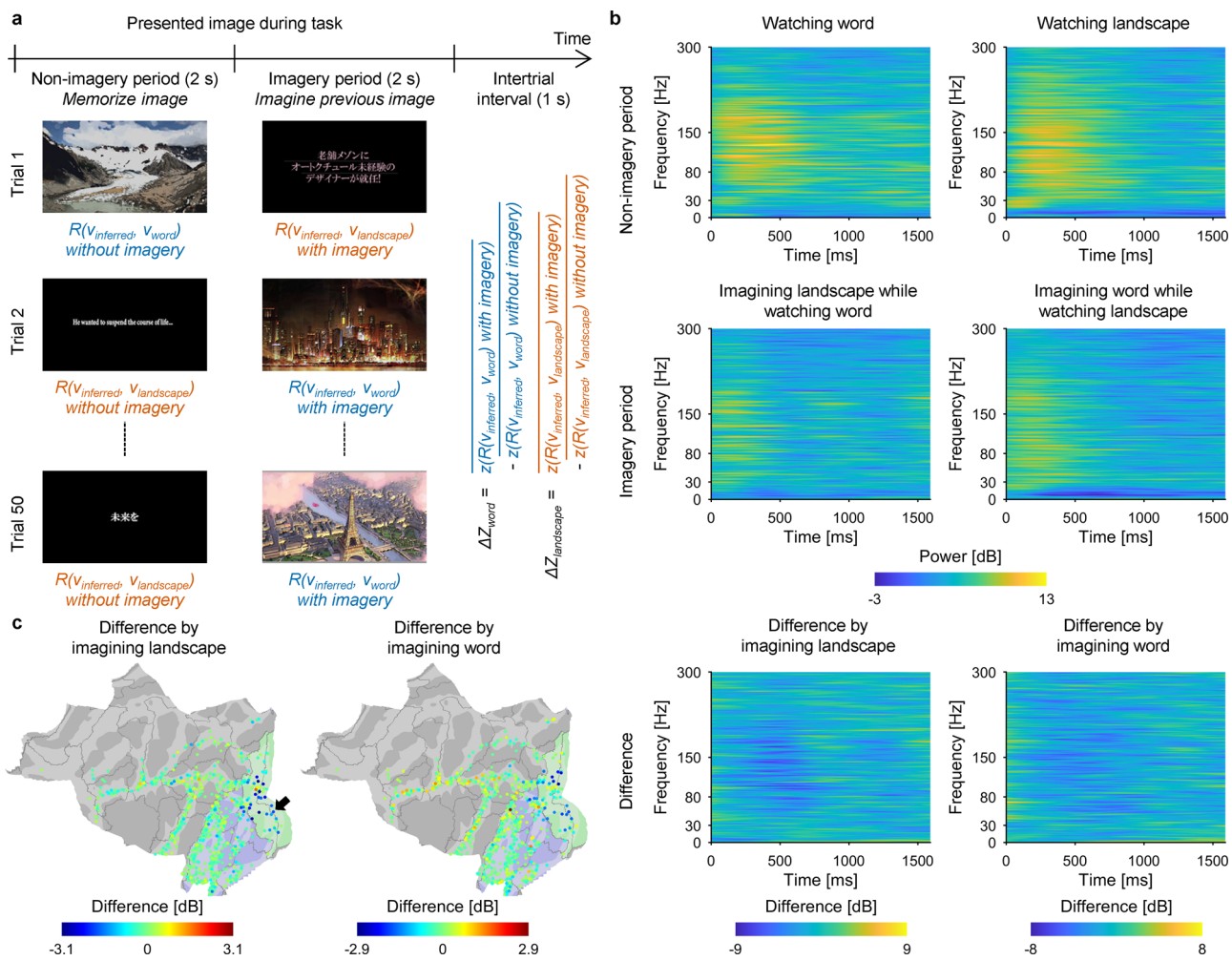

**Fig. 6 ECoGs during the imagery task. a** Schematic of the imagery task. After the presentation of the word or landscape image (nonimagery period), the subjects imagined the word image during the presentation of the landscape image, or imagined the landscape image during the presentation of the word image (imagery period). For all combinations of images, the trials were repeated in randomized order, resulting in 50 trials. To obtain modulation of $R(v_{inferred}, v_{word})$ due to imagining the word image ($\Delta Z_{word}$), the $R(v_{inferred}, v_{word})$ during the presentations of landscape images were Fisher z-transformed and averaged within the nonimagery period and the imagery period to calculate the difference between them $\overline{(z(R(v_{inferred}, v_{word}))}$ during imagery period – $\overline{z(R(v_{inferred}, v_{word}))}$ during nonimagery period). Similarly, $\Delta Z_{landscape}$ was evaluated from the $R(v_{inferred}, v_{landscape})$ during the presentation of word images. **b** Results of time–frequency decomposition for the ECoGs of the electrode indicated by the black arrow in (**c**). The upper and middle panels represent the time–frequency maps of ECoGs while watching the image of words (left) and landscapes (right) during the nonimagery period and imagery period, respectively. The difference between these two maps of each column is shown in the bottom panel. **c** Powers in the high-γ band from 0 to 1 s after the presentation of word images (left) and landscape images (right) were subtracted between the two periods (imagery period – nonimagery period) to be shown on the cortical surface at the location of each subdural electrode with color-coding.

control only by attention. Among the 2926 images used for the feedback, the human face was a common attribute. In fact, 1691 images (57.8%) contained the human face attribute; nevertheless, it was difficult to display the images of the human face as instructed. Moreover, the significant modulation in the imagery task, in which no images contained both meanings, suggested that the neural representation was modulated by the imagery, not by the attention to a part of an image. Notably, it might be possible to use various strategies other than imagery and attention—for example, internal verbalization—to control the inferred vector. However, because we trained the decoder using the ECoGs while the subjects were simply watching videos, it is unlikely that the trained decoder responded to mental strategies other than vision-related strategies. Our results suggested that real-time feedback was controlled by both attention and mental imagery, which modulated the information inferred from the visual areas.

Little is known about how the perception and imagery share their neural representations, although their activated areas are known to largely overlap[25]. In the present study's real-time feedback task, the decoder trained using neural activity during perception showed significant control for a part of the semantic space (#1, #2, #10, and #15 of 14 significant principal components in the video-watching task). Although interpreting the meaning of the components (except #1 and #2) was difficult (Supplementary Table 2), the results were consistent with the previous study demonstrating that only a subset of the features of the imagined images (e.g., emotional features) are encoded in brain activity when images are imagined than when they are perceived, depending on the imagined category[26]. Interestingly, the previous study also suggested that some of those features activate neural representation differently during perception and imagery, which is in line with our result showing negative correlation during the real-time feedback task (#6 in Fig. 5f). In addition, the imagery

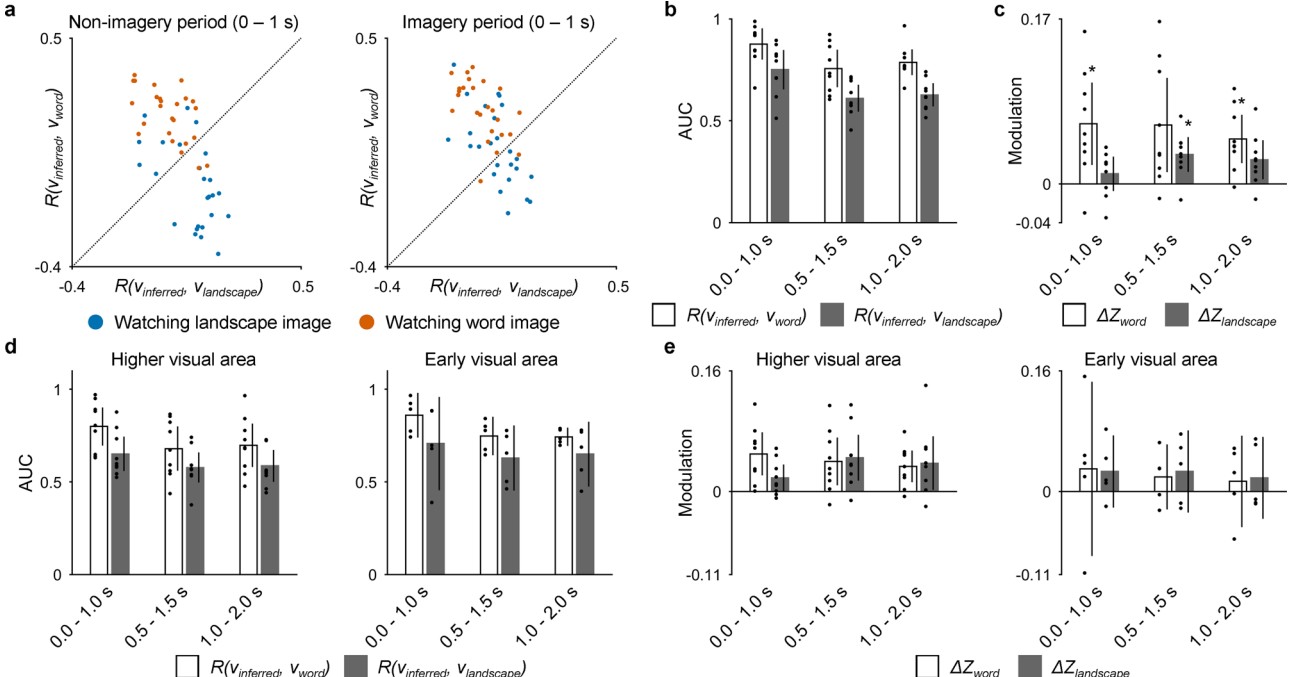

**Fig. 7 Modulation of the inferred semantic vectors by mental imagery. a** The $R(v_{inferred}, v_{word})$ (vertical axis) and the $R(v_{inferred}, v_{landscape})$ (horizontal axis) for each image were plotted with color-coded (blue, presentation of landscape image; red, presentation of word image) points for E01. **b** For each 1-s time window, the AUC to identify the category of the presented image (word or landscape) in the nonimagery period solely using $R(v_{inferred}, v_{word})$ or $R(v_{inferred}, v_{landscape})$ is shown with white and black bars, respectively. Individual values are shown with dots. Error bars denote the 95% CIs among subjects ($n = 9$). **c** $\Delta Z_{word}$ and $\Delta Z_{landscape}$ averaged across the subjects ($n = 9$) are shown with white and black bars, respectively, for each 1-s time window. Individual values are shown with dots. Error bars denote 95% CIs among subjects. *$P < 0.0083$ (Bonferroni-adjusted α-level; 0.05/6), one-sided one-sample t-test. **d** For each of three 1-s time windows in the nonimagery period, the AUC to identify the category of the presented image based on $R(v_{inferred}, v_{word})$ or $R(v_{inferred}, v_{landscape})$ is shown from using the high-γ features from the higher visual area (left; $n = 9$) and early visual area (right; $n = 5$). Individual values are shown with dots. Error bars denote 95% CIs among subjects. **e** For each 1-s time window and for the high-γ features from higher visual area (left; $n = 9$) and early visual area (right; $n = 5$), $\Delta Z_{word}$ and $\Delta Z_{landscape}$ averaged across the subjects are shown with white and black bars, respectively. Individual values are shown with dots. Error bars denote 95% CIs among subjects.

task revealed that modulation of the inferred vector by imagery was asymmetrical between word and landscape; $\Delta Z_{word}$ was the largest in the early time period of the imagery, whereas $\Delta Z_{landscape}$ became significant at 1 s after initiation of the imagery. Moreover, the modulation might depend on the anatomical locations of the implanted electrodes, being greater in the higher visual area than in the early visual area. This observation is consistent with findings in a previous study that, in the ventral stream, the higher visual areas rather than the early visual areas more similarly represent imagery and perception[15], because similar representation makes the perception decoder controllable by imagery. Lastly, the temporal dynamics of the shared neural representation are interesting. Previous studies showed that θ- and γ-activity convey the bottom-up information, and the α- and β-activity convey top-down signals[27,28]. In line with these studies, our results showed the highest decoding accuracy in the high-γ band (Supplementary Fig. 3e). Moreover, the imagery task revealed that the neural representation in the high-γ band was modulated by the imagery, suggesting shared neural representation between perception and imagery in that band. The suggested explanation is that coupling of the high-γ activity caused by a visual stimulus with a lower frequency band such as α[29] or β[30] serves as top-down control for stimulus processing. On the other hand, a recent study using electroencephalograms reported shared neural representations in the α band[31]. Further studies are necessary to reveal how the bottom-up information represented by the spatiotemporal pattern of cortical activities is modulated by top-down signals.

This study was characterized by the use of semantic space to test the hypothesis. We constructed the semantic space based on the skip-gram model using annotations by humans, which were consistent across annotators (Supplementary Fig. 2d), and confirmed that the semantic vectors of the images represented the semantic attributes of images such as word, landscape, and human face (Fig. 2b). Moreover, it was shown that the high-γ features in the visual areas while watching the training videos were significantly inferred from the semantic vectors even when excluding the contribution of the low-level visual and auditory features (Fig. 2d). These results demonstrate that the semantic vectors used in our study contained the semantic information of visual stimuli and succeeded in explaining the visual cortical responses for various semantic attributes. Compared with previous studies demonstrating such semantic representation in ECoGs[19,32,33], our study enrolled a large number of subjects and various visual stimuli to reveal semantic representations in the ECoGs.

It should be noted that the semantic space might affect the controllability of the rBCI. Semantic space can be based on semantic categories by human judgment[34–36] or automatically learned from a large text corpus by language models[20,37]. Interestingly, both methods extract similar spaces[38]. Moreover, recent studies have demonstrated that intermediate layers of deep neural networks are applicable for decoding visual stimuli[5]. It should not be forgotten that the best space for decoding might differ from that for the control of rBCI. The decoding accuracy results in the video-watching task was higher for the first principal component

than for the second principal component (Fig. 3a), but in the closed-loop condition, the opposite relation held (Fig. 5f). The controllability of the inferred image did not seem to depend on the decoding accuracy in the video-watching task but might depend on the semantic attributes. The difference in controllability based on the target category could not be evaluated because of the decoding scheme (Supplementary Fig. 5c); however, our result seems consistent with previous studies suggesting that the high accuracy of identifying perceiving images by perception decoders does not guarantee high accuracy in identifying mental imagery by the decoder[39,40]. The optimum space with the best controllability for the rBCI should be investigated in further studies under closed-loop conditions.

Compared with the condition without feedback, the closed-loop condition might improve the accuracy with which the imagined category could be inferred. A previous study demonstrated that a decoder to infer perceived visual stimuli can identify some objects imagined by eyes-closed subjects; however, the accuracy to identify the imagined objects was low compared to the accuracy to identify the perceived objects[10]. Consistently in the closed-loop condition, the accuracy to infer the imagined category was low at the beginning of each trial when the subject imagined images based on instruction, but the feedback screen was black (Fig. 5c and Supplementary Fig. 5a). However, as feedback continued, the subjects succeeded in controlling the online vector to be closer to the instructed semantic vectors with higher accuracy. The low accuracy at the beginning might be also explained by a too short duration to form vivid imagery; however, the decoders also failed to identify the category of the imagined images even in the later time windows of the imagery period in the imagery task (Supplementary Fig. 6c), in which significant modulations were observed. The suggestion is therefore that the accuracy of inferring the imagined category was improved in the closed-loop condition compared with the imagined category being decoded without feedback.

Last, rBCI might be useful as a communication device for severely paralyzed patients, such as those with amyotrophic lateral sclerosis (ALS)[41], to display patients' thoughts as images. For these individuals, a brain-computer interface (BCI)[42] is in high demand[43] and succeeds in expressing their thoughts by controlling some communication tools[16,44–46], although most BCIs rely on motor-related activities, which degenerate in patients with ALS. Because visual cortical activity persists for a long time[47] even in patients with ALS, rBCI using visual cortical activity might be used as a stable communication device for patients with severe ALS[48]. Further study of rBCI will allow the development of novel communication devices for severely paralyzed patients.

## Methods

**Subjects**. This study included a total of 21 subjects with drug-resistant epilepsy (14 males; 25.0 ± 11.2 years old, mean ± standard deviation [SD] on the day of the experiment) from three sites. One subject participated twice because of a second surgery after an interval of 2 years (E07 and E11). The subjects were implanted with intracranial electrodes prior to the study for the purpose of treating their epilepsy (number of subdural electrodes: 64.4 ± 17.0; the number of depth electrodes: 10.8 ± 10.0). All participants were recruited from patients implanted with electrodes at three university sites (Osaka University, Juntendo University, and Nara Medical University). Participants were recruited based on recommendations from the surgeons who placed the electrodes. Prior to the experiment, written consent was obtained from all subjects after explaining the nature and possible consequences of the study. The experiment was performed in accordance with the experimental protocol approved by the ethics committee of each hospital (Osaka University Medical Hospital: Approval No. 14353, UMIN000017900; Juntendo University Hospital: Approval No. 18-164; Nara Medical University Hospital: Approval No. 2098).

**Sample size**. The amount of data collected per patient depended on their clinical treatment schedule and the amount of time each participant was willing to volunteer for the study. According to our previous work[33], the duration of the training videos in the video-watching task and the number of trials in the real-time feedback and imagery tasks were determined to be sufficient to train a decoder and to show the modulation of the neural activities in their corresponding analyses, respectively. No data were excluded from the analysis. Reproducibility of the control of semantic vectors in the real-time feedback task was confirmed with four independent study participants.

**Experimental settings and ECoG recordings**. The subjects either sat on beds in their hospital rooms or were seated on chairs to perform the experimental tasks. A computer screen was placed in front of the subjects to show the video stimuli, the real-time feedback image, or stimuli for the imagery task. During the experiment, ECoGs were recorded at 10 kHz by EEG-1200 (Nihon Koden, Tokyo, Japan) by referencing the average of two intracranial electrodes. The presentation timing of the visual stimuli and real-time feedback images was monitored by DATAPixx3 (VPixx Technologies, Quebec, Canada) such that the digital pulse at the timing of the presentation was recorded synchronously with the ECoG. Gaze data were also recorded using an eye-tracking system (Tobii, Danderyd, Sweden) to monitor if the subjects were performing tasks, with the exception of those who wore glasses for vision correction or those in whom the position of wirings to the intracranial electrodes interfered with the system.

**Experimental procedures**. Seventeen subjects (E01–E17) participated in the video-watching task to evaluate the relationship between the semantic vector of each scene and ECoGs while watching the scenes. Four subjects (E01–E04) participated in the real-time feedback task, in which a decoder trained with ECoGs recorded during the video-watching task was used to determine feedback images. Moreover, 13 subjects (E01–E09 and E018–E21), including the four subjects from the real-time feedback task, participated in the imagery task to elucidate modulation of inferred semantic vectors (output of the decoders) by visual mental imagery.

**Video-watching task: task procedures**. While ECoGs were recorded, 17 subjects (E01–E17) watched the six 10-min videos (training videos). Among them, 12 subjects (E01, E03, E06, E07, and E09–E16) watched a 10-min video (validation video) composed of four repetitions of a 2.5-min movie. ECoGs of baseline brain activity were recorded prior to the presentation of each video using one of the following two methods. (1) Thirteen subjects (E05–E17) were instructed not to think of anything or move or to remain calm for 30 s (resting without images). (2) The remaining four subjects (E01–E04) were presented with sixty images for 1 s each and subsequently participated in the real-time feedback task; the subjects were instructed to watch the images by keeping their eyes on the red fixation point at the center of the images without thinking of anything or moving while remaining calm (resting with 60 images). The images were selected from ImageNet[49] and cropped at their center to create square images; the order of the images was randomized for each video. In addition, ECoGs were recorded during a 30-s resting period (which is the same condition as in method 1).

Immediately after recording the baseline brain activity, one of the 10-min videos was presented to the subject with audio of the video played from a pair of speakers. No fixation point was presented during the video, and the subjects were instructed to watch the video freely. To minimize the subject's fatigue, some interval was taken between the presentations of the six training videos; consequently, the entire task for the training videos took 1–3 days to complete. The validation video was presented after the presentation of training videos.

**Video-watching task: videos for visual stimuli**. We created the training videos and the validation video composed of 224 and 44 short cinema or animation clips, respectively, each of which was cut out from one of 75 trailers or behind-the-scene features downloaded from Vimeo. Those trailer or behind-the-scene features originated from 70 video sources (cinema or animation). The median duration of the clips was 16 s (interquartile range, 14–18 s), and they were sequentially concatenated to create six 10-min videos as training videos and a 2.5-min video to be repeated four times, resulting in a 10-min validation video. The videos contained scenes that varied widely in semantic content, such as scenery of nature, space, animals, food, people, and text. No overlapping scenes were included in the videos except the repetitions in the validation video; it should be noted that the training videos and the validation videos did not have any overlap although they originated from the same trailers or behind-the-scenes features.

**Video-watching task: construction of the skip-gram model**. A skip-gram model was trained using the Japanese Wikipedia dump data following the procedure described in a study by Nishida and Nishimoto[7]. The Japanese text of the articles in the Wikipedia dump was segmented into words and lemmatized to create a text corpus. This conversion was performed using MeCab[50], an open-source text segmentation software, and the Nara Institute of Science and Technology (NAIST) Japanese dictionary, a vocabulary database for MeCab. Words other than nouns, verbs, and adjectives were discarded from the text corpus, in addition to those that appeared fewer than 120 times. After these pre-processing steps, the text corpus had 365,312,470 words, including 94,337 nouns, 4922 verbs, and 631 adjectives. A skip-gram model was trained with the text corpus using the Gensim Python library

with the following parameters: the dimension of word vector representation, 1000; window size, 5; number of negative samples, 5; use of hierarchical Softmax, no. For presentation in this article, Japanese words (i.e., annotations and the instructions in the real-time feedback task) were translated to English using Google Translate.

**Video-watching task: construction of the semantic vector**. From the six 10-min training videos and the 2.5-min movie in the validation video, a still image of the video scene was extracted every second (Supplementary Fig. 2c). Each image (scene) was manually annotated by five annotators with descriptive sentences that had 50 or more Japanese characters. The annotators were native Japanese speakers who were neither authors nor subjects. Semantic vectors for each scene were constructed based on the vector representations learned by the skip-gram model[20], which enables linear operation between vectors representing words (e.g., vector operation of "king" – "man" + "woman" results close to "queen"[51,52]. All annotations were first segmented into words and lemmatized using MeCab and the NAIST Japanese dictionary. Among the lemmatized words in the annotations, words that did not exist in the text corpus were discarded. Each word was converted into the corresponding 1000-dimensional vector representation using the trained skip-gram model. Based on the linear relationships between semantic vectors, semantic vectors were constructed by averaging the vector representations of the lemmatized word, first within each annotation, and then across the five annotations for each scene (denoted as $V_{true} := \{v^i_{true} | i = 1, \cdots, 3600 \text{(scene)}\}$ for the training videos). It should be noted that the average of the correlation coefficients across vectors for the five annotators for the same scene was $0.7523 \pm 0.0013$ (mean ± 95% CIs among the scenes), suggesting the high consistency of the vectors across the annotators.

**Video-watching task: evaluation of the visual semantic space**. To visualize the space spanned by the semantic vectors, PCA was applied to the 3600 semantic vectors ($V_{true}$) from the training videos to reveal their major components. To create the vector representation for "human face" ($v_{face}$), the vectors for "human" and "face", which were learned by the skip-gram model, were averaged. For the vector representation for "word" and "landscape", the corresponding vectors in the skip-gram model ($v_{word}$ and $v_{landscape}$) were used.

**Video-watching task: extraction of low-level visual and auditory features**. Low-level visual and auditory features were extracted by applying motion energy filters[8,22] to the training videos and modulation-transfer function models[23] to the sound of the training videos, respectively. To extract low-level visual features, the videos were down-sampled in frame rate and in spatial resolution to create videos with 15 fps and $171 \times 96$ pixels of RGB colors. The videos were then cropped at their center and converted to Commission Internationale de l'Éclairage $L^*A^*B^*$ color space. Motion energies were acquired from the luminance of the videos by applying spatiotemporal Gabor filters differing in motion direction (0, 45,···, 315°; orientation of each filter was perpendicular to the motion direction), spatial frequency (0, 1.5, 3, 6, 12, and 24 cycles/image), and temporal frequency (0, 2, and 4 Hz), spatially positioned on a square grid with a distance of 4.0 SDs of the spatial Gaussian envelope. By averaging the log-transformed motion energies within the 1-s time window corresponding to each scene, low-level visual features were calculated as 2139-dimensional vectors. Meanwhile, the low-level auditory features were extracted from the sound of the videos using modulation-transfer function models[53]. The sound of the videos was converted to a spectrogram using 128 bandpass filters with a window size of 25 ms at a step of 10 ms. By applying 100 modulation-selective filters (10 spectral modulation scales and 10 temporal modulation rates) to the spectrogram, modulation energies were calculated. The modulation energies were then log-transformed and averaged within the 1-s time window corresponding to each scene and within 20 nonoverlapping frequency bands evenly spaced in log-space from 20 Hz to 10 kHz, resulting in 2000-dimensional vectors of low-level auditory features.

**Real-time feedback task: task procedures**. Four subjects (E01–E04) participated in the real-time feedback task that was conducted on a different day from the video-watching task. The subjects were first informed that the images would be presented based on their real-time brain activity and were instructed to control the feedback image on the screen by visual imagery so that the feedback image keeps showing the instructed category (Fig. 1b).

The real-time feedback task was composed of four sessions, each consisting of 30 trials. Prior to the first session, ECoGs of the baseline brain activity were recorded while subjects watched the same 60 images that had been presented before the video-watching task (i.e., resting with 60 images). Each image was presented for 1 s in randomized order without intervals. During this period, the subjects were instructed to watch the images by keeping their eyes on the red fixation point at the center of the images without thinking of anything or moving while remaining calm.

On each trial, a black screen was first presented for 4.5 s; 2.5 s after display of the black screen, one of the following three instructions (target categories) whose durations were less than 1.0 s was given orally in Japanese: "Moji" (word), "Fuukei" (landscape), or "Hito-no-kao" (human face). After the presentation of the black screen, 32 frames of feedback images were presented, each with a duration of

250 ms (Fig. 4a). The feedback image shown on the screen was one of 2926 images out of 3600 annotated images in the training videos. The other 674 images were discarded because they were blurry or otherwise unclear or because they contained text that might evoke negative feelings (e.g., "death").

**Real-time feedback task: real-time decoding**. In the real-time feedback task, ECoGs were acquired and decoded in real time to infer a semantic vector. The ECoGs of the most recent 1 s were re-referenced by pre-processing, converted into raw features, and compensated with the re-referenced baseline ECoGs from the resting with 60 images condition recorded just before the real-time feedback task (for details, see "Signal pre-processing" and "Calculation of high-γ features"; for electrodes used in the real-time decoding, see Fig. 4b). Then, the semantic vector was inferred from the compensated decoding features for the feedback. The feedback image was determined based on the highest Pearson's correlation coefficient between the online vector ($v_{online}$) and the true semantic vectors of the 2926 scenes. The first $v_{online}$, used to determine the first feedback image in each trial, was the inferred vector ($v_{inferred}$) from the 1-s ECoGs during the presentation of the black screen. The subsequent $v_{online}$ was calculated as the linear interpolation of the previous online vector ($v^{prev}_{online}$) and the inferred vector of the most recent 1-s ECoGs, which were acquired at 250-ms intervals with the previous decoding ($v_{online} = \alpha \cdot v_{inferred} + (1 - \alpha) \cdot v^{prev}_{online}$). The interpolation weight ($\alpha$) was manually adjusted prior to the first real-time feedback session by experimenters and fixed throughout all sessions (E01 and E04, $\alpha = 0.5$; E02, $\alpha = 1.0$; E03, $\alpha = 0.4$). Within each real-time feedback session, each instruction was given ten times in randomized order, and the sessions were repeated four times for each subject at such an interval that the subject could take a break. The system delay from the acquisition of ECoGs to the presentation of the feedback image was $195.2 \pm 29.2$ ms (mean ± SD; measured from the real-time sessions with E01).

**Imagery task: task procedure**. The subjects visually imagined mental images in one category while watching various images in another category (Fig. 6a). The images shown in this task originated from the 3600 annotated images of the training videos. We selected five images for both the word and landscape categories based on the highest Pearson's correlation coefficients between the true semantic vector and the semantic vector for word and landscape ($R(v^i_{true}, v_{word})$ and $R(v^i_{true}, v_{landscape})$), although some images were rejected such that no images included meanings related to a different category, and the selected images had a clear meaning of either word or landscape (for selected images, see Supplementary Fig. 6a). At the beginning of the task, baseline ECoGs were acquired for 30 s, during which period the subject was instructed not to think of anything or move and to remain calm (resting without images). Then, on each trial, the first image was presented for 2 s (nonimagery period), and the second image, selected from a different category, was presented for another 2 s (imagery period). No fixation points were shown during the presentation of these images. The intertrial interval was 1 s, and the subject was presented with a black screen with a white cross at its center during this period. For all combinations of images between the two categories, the trials were repeated; hence, the number of trials for one session was 50. The presentation order was randomized. The subjects were instructed to memorize the first image and then to visually imagine the memorized image while watching the second image. Twelve subjects (E01–E08 and E18–E21) participated in the imagery task for one session, whereas E09 participated in two sessions.

**Signal pre-processing**. Based on visual inspection of the recorded ECoGs in the video-watching task (E01–E17) or in the imagery task (E18–E21), noisy channels were discarded from subsequent analyses. Neither down-sampling nor filtering was performed as signal pre-processing.

**Signal pre-processing: video-watching task and imagery task**. ECoGs obtained during the video-watching task and the imagery task were re-referenced by common averaging across the noise-free channels.

**Signal pre-processing: real-time feedback task**. During the real-time decoding (and for the training of the decoder used in the real-time feedback task), a subset of the noise-free channels was used to re-reference the ECoGs to increase performance in the real-time feedback task. (1) We first discarded channels in regions that were considered to not contribute to the control of the inferred vector (e.g., channel whose electrode was located in the frontal lobe). (2) Channels that did not show a stable response to the visual stimuli were discarded as follows: (2–1) ECoGs from the video-watching task were re-referenced using common averaging across the remaining channels. (2–2) From the re-referenced signals, raw features during the presentation of the images in the resting with 60 images condition were calculated to discard channels that satisfied the following criteria in at least one video: $standard\ deviation(feature^{baseline,\ raw}_i / feature^{baseline0,\ raw}_i) > 0.5$ (see "Calculation of high-γ features"). (2–3) The ECoGs in the video-watching task were again re-referenced using common averaging across the remaining channels to train the decoder. Re-referencing using the same subset of the noise-free channels was performed during the real-time decoding and in the analysis of the ECoGs

recorded in the real-time feedback task (for locations of selected electrodes, see Fig. 4b).

**Calculation of powers for consistency analysis.** For the consistency analysis, powers were calculated from 1-s ECoGs of a channel ($X_{signal}(t)$) while the subjects were watching the validation video. Using a Hamming window and fast Fourier transformation (FFT), the power spectrum density was calculated ($PSD_f(X_{signal}(t)) [\mu V^2/Hz]$, $f$ : frequency [Hz]) to be averaged within the α (8–13 Hz), β (13–30 Hz), low-γ (30–80 Hz), and high-γ (80–150 Hz) frequency bands.

**Calculation of high-γ features.** From 1-s ECoGs of a channel ($X_{signal}(t)$) and the corresponding baseline signals from the same channel ($X_{baseline}(t)$), decoding features were calculated. The power spectrum density of the signal was calculated using a Hamming window and FFT; the power spectrum density was then averaged across the high-γ frequency band to calculate the raw features as follows:
$$feature^{signal, raw} := \left(\frac{1}{|\{f \in [80,150]\}|} \sum_{f \in [80,150]} PSD_f(X_{signal}(t))\right)^{0.5} [\mu V/Hz^{0.5}].$$

To compensate for impedance changes for each intracranial electrode from the condition during the first training video presentation in the video-watching task, we evaluated the compensation factor ($comp$) to calculate decoding features ($feature^{signal} = feature^{signal, raw}/comp$) using one of the following two methods, each of which corresponded to the recording method for the baseline signals.

(1) Resting without images
The ECoGs during the 30-s resting baseline ($X_{baseline}(t)$) were divided into 30 time windows of 1 s each. For each time window, the raw feature ($feature_i^{baseline, raw}$, $i = 1, \cdots, 30$ (time window)) was calculated. The same procedure was applied to the ECoGs during the 30-s resting baseline recorded just before the presentation of the first training video ($X_{baseline0}(t)$) to acquire raw features ($feature_i^{baseline0, raw}$). The compensation factor was defined as follows: $comp := \sum_i feature_i^{baseline, raw}/\sum_i feature_i^{baseline0, raw}$.

(2) Resting with 60 images
The ECoGs from 0 to 1 s after the presentation of each of the 60 images were extracted from the baseline ECoGs ($X_{baseline}(t)$) to obtain raw features ($feature_i^{baseline, raw}$, $i = 1, \cdots, 60$ (image)). In the same manner, from the ECoGs recorded prior to presentation of the first training video ($X_{baseline0}(t)$), the signals from 0 to 1 s after presentation of the images were extracted and converted to the raw features ($feature_i^{baseline0, raw}$). By paring raw features for the same image, the compensation factor was defined as follows: $comp := \sum_i (feature_i^{baseline, raw}/feature_i^{baseline0, raw})/60$.

**Construction of the decoder.** Throughout this study, ridge regression was used to infer semantic vectors from decoding features. To train a regression model from a dataset that consisted of decoding features and corresponding true semantic vectors, cross-validation[54] was applied; the decoding features in the training dataset of each fold were standardized by z-scoring using its mean and SD for each dimension of the features. The decoding features in the testing dataset were standardized using the same means and SDs calculated solely with the training dataset to prevent any data leakage[55]. For each candidate of the regularization parameter of the ridge regression ($10^{-8}, 10^{-7}, \cdots, 10^8$), a regression model was trained for each dimension of the visual semantic space to decode the standardized decoding features of the testing dataset. Finally, all decoding features in the entire dataset were again standardized by z-scoring, using the mean and SD for each dimension of the features; a ridge regression model was trained for each dimension of the visual semantic space with the regularization parameter that maximized the average of the dimension-wise correlation coefficients in the visual semantic space ($\{DimR^k(V_{inferred}, V_{true})\}$; see "Measure of accuracy for the inferred semantic vectors") between the sequence of the inferred semantic vector ($V_{inferred} := \{v_{inferred}^i | i$ : scene in entire dataset$\}$) and the true semantic vector ($V_{true} := \{v_{true}^i\}$). When the trained decoder was applied to new decoding features, the new decoding features were standardized by the same means and SDs of the entire dataset before applying the regression models.

**Measure of accuracy for the inferred semantic vectors.** To evaluate the decoding accuracy of the semantic vectors inferred from the ECoGs, we used the following four measures.

(1) Dimension-wise correlation coefficients ($DimR^k$)
Dimension-wise correlation coefficients were calculated as a Pearson's correlation coefficient between sequences of the true semantic vectors ($V_{true} := \{v_{true}^i | i$ : tested scene$\}$) and the inferred semantic vectors ($V_{inferred} := \{v_{inferred}^i\}$) for each dimension ($\{DimR^k(V_{inferred}, V_{true}) | k = 1, \cdots, 1000$ (dimension in the visual semantic space)$\}$).

(2) Projected correlation coefficients ($PrjR^k$)
Projected correlation coefficients ($\{PrjR^k(V_{inferred}, V_{true}) | k = 1, \cdots, 1000\}$) were measured using the direction vectors of the true semantic vectors acquired by PCA for the training videos (see "Video-watching task:

evaluation of the visual semantic space"). Each inferred semantic vector from the tested scenes ($V_{inferred} := \{v_{inferred}^i | i$ : tested scene$\}$) was projected to the $k$th direction vector of the PCA to calculate Pearson's correlation coefficients between the projected values and the $k$th principal component of the corresponding true semantic vectors ($V_{true} := \{v_{true}^i\}$).

(3) Scene-wise correlation coefficient ($R(v_{inferred}^i, v_{true}^i)$)
The scene-wise correlation coefficient was calculated to evaluate the accuracy for each image. The scene-wise correlation coefficient was defined as the Pearson's correlation coefficient between the true and inferred semantic vectors for each scene ($R(v_{inferred}^i, v_{true}^i)$).

(4) Scene-identification accuracy
Scene-identification accuracy indicated the accuracy in identifying the corresponding scene from the inferred semantic vector. For a scene ($i$) to be tested, the true scene-wise correlation coefficient ($R(v_{inferred}^i, v_{true}^i)$) between the inferred semantic vector ($v_{inferred}^i$) and the true semantic vector ($v_{true}^i$) was compared in a pairwise manner with other scene-wise correlation coefficients between the inferred semantic vector and true semantic vectors of other scenes to be compared ($\{v_{true}^j | i \neq j, j$ : scene to be compared$\}$). The proportion of compared scenes with which the inferred semantic vector correlated less than or equal to the true scene-wise correlation coefficient was calculated, and the average of the proportions was defined as the scene-identification accuracy of the tested scene ($acc^i := \sum_j (C_{i,j})/|\{j\}|$ where $C_{i,j} := 1$ if $R(v_{inferred}^i, v_{true}^i) > R(v_{inferred}^i, v_{true}^j)$ otherwise 0).

**Analysis for the video-watching task: consistency analysis of cortical activity while watching the validation video.** To determine the frequency band that responded to the visual stimulus most consistently, the consistency of the cortical activity was evaluated using the four repetitions in the validation video. For each electrode, powers in four frequency bands (α, β, low γ, and high γ) were calculated from 1-s ECoGs obtained without overlap (150 values for each electrode, band, and repetition). For all six possible pairs among the four repetitions, Pearson's correlation coefficients were calculated between the powers; the correlation coefficients were then Fisher z-transformed to be averaged across the pairs.

**Analysis for the video-watching task: extraction of high-γ features.** For further encoding and decoding analyses, high-γ features were extracted from pre-processed ECoGs recorded during the video-watching task. For each annotated image in the videos, ECoGs within a time window of ±500 ms combined with baseline ECoGs recorded prior to each video presentation were used to calculate the features. For the four subjects who participated in the real-time feedback task (E01–E04), the baseline ECoGs from the resting with 60 images condition were used for compensation to evaluate the decoding performance during the video-watching task in a condition closer to the real-time decoding task; for the other 13 subjects (E05–E17), baseline ECoGs from the resting without images condition were used for compensation.

**Analysis for the video-watching task: division of scenes for nested cross-validation.** In further encoding and decoding analyses, tenfold nested cross-validation was applied to enable accurate evaluation; hence, scenes in the training videos were divided into ten groups without any overlapping scenes between them. To prevent overestimation of the accuracy that might be caused when similar scenes from the same video source were divided into different groups, the division of the scenes was determined using a generic algorithm such that (1) scenes from the same video source were kept in the same group and (2) the imbalance of the number of scenes in each group was minimal.

**Analysis for the video-watching task: inference of the high-γ features from the semantic vectors and the low-level visual and auditory features (encoding analysis).** To reveal how the high-γ features responded to the videos, the high-γ features while subjects watched the training videos were inferred from the semantic vectors and the low-level visual and auditory features of the videos using ridge regression for each electrode. Low-level visual and auditory features were obtained by applying motion energy filters[8,22] to the training video and modulation-transfer function models[23] to the sound of the videos (see "Video-watching task: extraction of low-level visual and auditory features"). The semantic features (semantic vectors) and low-level visual and auditory features were all concatenated to infer high-γ features at each electrode using tenfold nested cross-validation in the following procedure. Based on the division of the scenes (see "Analysis for the video-watching task: division of scenes for nested cross-validation"), the dataset, consisting of the concatenated features and the high-γ features, was divided into ten smaller datasets. As described in Construction of the decoder, for each dataset (test dataset), a regression model was trained using samples from all other datasets (training dataset) to infer the test dataset. In this way, the regularization parameter was selected without any over-fitting[55]. To evaluate the contributions of each feature set (semantic features, low-level visual features, or low-level auditory features), each test dataset was inferred with the regression model whose weights for the other two feature sets were set to zero.

**Analysis for the video-watching task: inference of the semantic vectors from the high-γ features in the video-watching task (decoding analysis).** Semantic vectors corresponding to all 3600 scenes of the training videos were inferred from the high-γ features using tenfold nested cross-validation. The dataset, consisting of the decoding features (high-γ features) and true semantic vectors, was divided into ten smaller datasets without any overlapping samples between them (see "Analysis for the video-watching task: division of scenes for nested cross-validation"). For each dataset (test dataset), a decoder was trained using samples from all other datasets (training dataset; see "Construction of the decoder") to decode the test dataset so that the regularization parameter was selected without any over-fitting[55]. To assess the decoding accuracy in higher and early visual areas, the decoding features from subdural electrodes in each area were decoded with the same procedure.

**Analysis for the video-watching task: evaluation of binary accuracy and accuracy among three categories.** Decoding performance in the video-watching task was evaluated based on the Pearson's correlation coefficients (scene-wise correlation coefficients) of the inferred semantic vector ($v_{inferred}$) with the semantic vectors of the categories ($R(v_{category}, v_{inferred})$, where $v_{category}$ was $v_{word}$, $v_{landscape}$, or $v_{face}$) (1) for the three categories and (2) for each pair of categories among the three categories (binary accuracy). The classification was considered correct when the $R(v_{category}, v_{inferred})$ was the highest for the category of the presented scene.

**Analysis for the real-time feedback task: construction of a decoder for real-time feedback.** The decoder used in the real-time feedback task was trained using the decoding features from all 3600 scenes of the six training videos presented in the video-watching task with the regularization parameter optimized by the cross-validation (for details, see "Signal pre-processing" and "Construction of the decoder"). The baseline ECoGs recorded in the resting with 60 images condition were used to compensate the decoding features. For the cross-validation used in the decoder training, the decoding features were divided such that those from the same recording day were treated as a group to maximize decoding performance on a different day from the measurement of the training data. Consequently, twofold cross-validation was applied for E03 and E04. Because E01 watched the training videos in one day, threefold cross-validation was used to group the scenes as being from the first and second videos, the third and fourth videos, and the fifth and sixth videos. The same threefold cross-validation was applied for E02, who watched one video on one day and the other five videos on a different day.

**Analysis for the real-time feedback task: evaluation of real-time feedback task.** Performance in the real-time feedback task was evaluated using the online vectors ($v_{online}^{i,j}$ where $i = 1, \cdots, 120$ (trial) and $j = 1, \cdots, 32$ (frame)), based on which the feedback images were selected, by the following three methods.

(1) By considering each trial of the real-time feedback task as a three-choice trial among the three instructions, the prediction accuracy of the target category was evaluated. For each frame in a trial, the Pearson's correlation coefficients of the online vector with the semantic vectors of the three categories ($v_{instruction}$: $v_{word}$, $v_{landscape}$, and $v_{face}$) were calculated. Then, the correlation coefficients were Fisher z-transformed and averaged across the 32 frames in the trial ($\overline{z(R(v_{online}^i, v_{instruction}))} := \sum_j z(R(v_{online}^{i,j}, v_{instruction}))/32$). If the (Fisher z-transformed and) frame-averaged correlation coefficient with the semantic vector of the target category (e.g., $\overline{z(R(v_{online}^i, v_{word}))}$ for target category of the word) was higher than the other two ($\overline{z(R(v_{online}^i, v_{word}))} > \overline{z(R(v_{online}^i, v_{landscape}))}$ and $\overline{z(R(v_{online}^i, v_{word}))} > \overline{z(R(v_{online}^i, v_{face}))}$), the prediction on that trial was considered to be correct.

(2) For all pairs of categories among the three categories, the same procedure as (1) was performed using 80 trials whose target categories were the selected categories.

(3) For the directions (acquired as the direction vectors by PCA) along which the semantic vectors were significantly inferred with positive projected correlation coefficients in the video-watching task, the projected correlation coefficient ($\{PrjR^k (V_{online}, V_{target}) \mid k$: significant direction$\}$) was evaluated between the online vectors of all frames of all trials concatenated ($V_{online} := \{v_{online}^{i,j}\}$) and the semantic vector of their corresponding target category ($V_{target} := \{v_{target}^{i,j}\}$).

**Analysis for the imagery task: time–frequency decomposition analysis.** To reveal how imagery affects ECoGs during perception in the time–frequency domain, time–frequency decomposition was performed on the ECoGs from the imagery task. For each subdural electrode, ECoGs were obtained for each image presentation in the nonimagery period and the imagery period, and the newtimef function in eeglab[56] with an FFT window size of 8192 was applied. For cortical mapping of the powers, the time–frequency map of the decomposed powers for each presentation was converted into decibels and averaged within the high-γ frequency band and 0 to 1 s.

**Analysis for the imagery task: evaluation of modulation of inferred semantic vectors by mental imagery.** To reveal the modulation of the inferred vectors that were attributable to mental imagery, the ECoGs in the imagery task were evaluated by decoding. This analysis was performed for the subjects who participated in both the video-watching task and the imagery task (E01–E09). For each of the first and second images in the imagery task, pre-processed ECoGs from 0 to 1 s after the presentation of images were extracted and converted to decoding features using the pre-processed ECoGs of the 30-s resting baseline (resting without images) recorded just prior to the imagery task as the compensation baseline. Decoding features from the subdural electrodes were decoded using a decoder trained in the same manner as the one used in the real-time decoding with the exception of the use of decoding features from the video-watching task compensated with the 30-s resting ECoGs recorded in the resting without images condition. With the inferred semantic vector ($v_{inferred}$), Pearson's correlation coefficients of the semantic vector for word ($R(v_{inferred}, v_{word})$) and landscape ($R(v_{inferred}, v_{landscape})$) were calculated. Distinguishability of the category of the presented image was evaluated by AUC based on these correlation coefficients with each semantic vector. The modulation of $R(v_{inferred}, v_{word})$ by imagining word images ($\Delta Z_{word}$) was evaluated as the difference in the Fisher z-transformed and averaged correlation coefficients with the semantic vector for word ($\overline{z(R(v_{inferred}, v_{word}))}$) in the second and first presentations of landscape images (i.e., in the imagery and nonimagery periods, respectively). Similarly, the modulation of $R(v_{inferred}, v_{landscape})$ by imagining landscape images ($\Delta Z_{landscape}$) was evaluated as the difference in the Fisher z-transformed and averaged correlation coefficients with the semantic vector for landscape ($\overline{z(R(v_{inferred}, v_{landscape}))}$) in the second and first presentations of word images. The same procedure above was repeated for ECoGs from 0.5 to 1.5 s and 1.0 to 2.0 s after image presentation and for the subdural electrodes in higher and early visual areas.

**Analysis for the imagery task: evaluation of binary accuracy during the nonimagery period.** Binary accuracy to predict the category of the images presented during the nonimagery period was calculated based on the inferred vectors used to evaluate the modulation. For each presentation of an image in a category, the true scene-wise correlation coefficient between the inferred semantic vector from 0 to 1 s after the image presentation and the true semantic vector of the category was compared with the scene-wise correlation coefficient between the inferred semantic vector and true semantic vectors of another category. The proportion of the images that showed higher correlation coefficients with the semantic vectors of corresponding categories was defined as the binary accuracy.

**Localization of intracranial electrodes.** Intracranial electrodes were located based on pre-surgical T1-weighted magnetic resonance images (MRIs) and post-surgical computed tomography (CT) images. The MRIs and CT images were acquired at the site where the electrodes had been surgically implanted. The scanners used to obtain the MRIs were the Discovery MR750 (GE Healthcare, Chicago, USA) for five subjects, the SIGNA Architect (GE Healthcare) for one subject, the Ingenia (Philips Healthcare, Amsterdam, Netherlands) for two subjects, the Achieva (Philips Healthcare) for five subjects, the MAGNETOM Prisma (Siemens, Munich, Germany) for three subjects, the MAGNETOM Skyra (Siemens) for four subjects, and the MAGNETOM Verio (Siemens) for one subject. The CT images were obtained using the Discovery CT750 HD (GE Healthcare) for five subjects, the SOMATOM Emotion 16 (Siemens) for three subjects, the Aquilion ONE (Toshiba Medical Systems, Tochigi, Japan) for four subjects, the Aquilion Precision (Toshiba Medical Systems) for three subjects, and the Aquilion PRIME (Toshiba Medical Systems) for six subjects. First, FreeSurfer[57] was used to extract the cortical surface for each subject from the MRIs. Then, BioImage Suite[58] was used to manually locate the positions of the intracranial electrodes from the CT images that were co-registered to the MRIs. Afterward, the positions of the subdural electrodes were mapped onto the cortical surface using the intracranial electrode visualization toolbox[59]. The cortical surface was registered to the surface of a template brain (fsaverage) to locate the positions of each electrode on the normalized brain (Fig. 1a and Supplementary Fig. 1). For the area-based analysis, each subdural electrode was assigned to one of 22 regions based on the parcellation of the human connectome project (vid. supplementary neuroanatomical results)[60]. Moreover, to locate the position of each electrode on the contralateral hemisphere of the template brain, another registration was performed on the template brain that was flipped left to right.

**Illustrations.** The graphics of the scenes in this paper were created by an illustration company (Medical Education, Tokyo, Japan) specialized in scientific illustration and were based on the annotated images from the videos and the annotations, such that the semantic meanings in the annotations were not lost in the resulting illustrations. Copyrights of the illustrations were transferred to the authors.

**Statistics and reproducibility.** The difference in the high-γ features while watching scenes corresponding to the three categories, each with 50 scenes, was tested with one-way ANOVA (Fig. 2c).

The correlation coefficients between the high-γ features and the inferred features while watching the training videos were tested by one-sided Pearson's correlation test (Fig. 2d). The one-sided test was applied because successful regression results in positive correlation coefficients.

The inferred semantic vectors using high-γ features while subjects were watching the training videos in the video-watching task were tested for each principal component of the PCA using a two-sided permutation test (Fig. 3a). The Fisher z-transformed and subject-averaged projected correlation coefficient $(z(PrjR^k(V_{inferred}, V_{true})))$ for each principal component were compared to a corresponding chance-level distribution. The chance distributions were created by (1) splitting the true semantic vectors based on the video clips, (2) shuffling their order, (3) concatenating them, and (4) calculating the subject-averaged projected correlation coefficient in exactly the same way. The permutation was performed 1 million times. The α-level was adjusted by Bonferroni correction for the number of components, which was 1000.

Binary accuracies of all subjects using high-γ features were tested against chance level (50%) using one-sided one-sample $t$-tests. Because the binary accuracy was expected to be higher than the chance level, a one-sided test was used. The α-level was adjusted by Bonferroni correction for the number of paired categories (three). The binary accuracy averaged across the three pairs was also tested against the chance level (50%) by one-sided one-sample $t$-tests with Bonferroni correction for the number of tests (higher visual area, early visual area, and all electrodes; Fig. 3b). Differences among the binary accuracies of the higher visual and early visual areas were tested by two-sided Welch's $t$-test (Fig. 3b).

Three-choice accuracy during the real-time feedback task was tested using a permutation test. Because the accuracy was expected to be higher than the chance level when the feedback image was under the control of the subjects, a one-sided permutation test was performed. The chance-level distribution of the three-choice accuracy was estimated by shuffling target categories of all trials 1 million times.

To find components that were significantly controlled during the real-time feedback task, the Fisher z-transformed and subject-averaged projected correlation coefficients $(\{z(PrjR^k(V_{online}, V_{target}))\})$ were evaluated using two-sided permutation tests (Fig. 5f). The permutations were performed 1 million times by shuffling the target categories of all trials. The α-level was adjusted by Bonferroni correction for the number of principal components that could be decoded significantly better than chance in the video-watching task.

For each semantic vector of word ($v_{word}$) and landscape ($v_{landscape}$), the distribution of Pearson's correlation coefficients between the inferred vectors ($v_{inferred}$) and the semantic vector was evaluated for its difference between the nonimagery and imagery periods. Because it was hypothesized that these correlation coefficients would increase during imagery, the Fisher z-transformed correlation coefficients for the nonimagery and imagery periods were tested using one-sided Welch's $t$-test (Fig. 7a and Supplementary Fig. 6b).

Among all nine subjects who participated in both the video-watching task and the imagery task, $\Delta Z_{word}$ and $\Delta Z_{landscape}$ using high-γ features were tested against no modulation (0). Because it was hypothesized that these modulations would become positive, one-sided one-sample $t$-tests were adopted for the test. The α-level was adjusted by Bonferroni correction for the number of tests (six) (Fig. 7c).

**Reporting summary**. Further information on research design is available in the Nature Research Reporting Summary linked to this article.

## Data availability

All data are available in the main text, the supplementary materials, or figshare (https://figshare.com/articles/dataset/Datasets_for_Fukuma_et_al_Communications_Biology/12916037)[61].

## Code availability

All analyses were performed using Matlab R2015b (Natick, MA, USA). Custom codes are also shared in figshare (https://figshare.com/articles/dataset/Datasets_for_Fukuma_et_al_Communications_Biology/12916037)[61].

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

## Acknowledgements

We thank all subjects for their participation. This research was conducted under the Japan Science and Technology Agency (JST) Core Research for Evolutional Science and Technology (JPMJCR18A5). This research was also supported in part by the JST Precursory Research for Embryonic Science and Technology (JPMJPR1506), Exploratory Research for Advanced Technology (JPMJER1801), Moonshot R&D (JPMJMS2012), Grants-in-Aid for Scientific Research from KAKENHI (JP26560467, JP17H06032, JP20K16466, JP15H05710, JP18H04085, JP20H05705, and JP18H05522), grants from the Japan Agency for Medical Research and Development (AMED) (19dm0207070h0001, 19dm0307103, 19de0107001, and 19dm0307008), and the Canon Foundation.

## Author contributions

Conceptualization: T.Y.; methodology: R.F. and T.Y.; investigation: R.F., T.Y., H.S., K.T., S.Y., Y.I., Y.F., S.O., N.T., and N.K.–M.; data curation, formal analysis, and software: R.F.; funding acquisition: R.F., T.Y., and H.K.; writing–original draft: R.F. and T.Y.; writing–review and editing: R.F., T.Y., S.N., Y.K., and H.K.; supervision: T.Y.

## Competing interests

The authors declare no competing interests.
