## [Peer Review File · Communications Biology]

This manuscript has been previously reviewed at another Nature Portfolio journal. This document only contains reviewer comments and rebuttal letters for versions considered at Communications Biology.

Reviewers' comments:

Reviewer #1 (Remarks to the Author):

I was brought into the review process after the first revision stage. Therefore, I deem it my task to check whether the current response to the previous reviewers is sound, rather than to raise new points. Nevertheless, I have read both the rebuttal and the manuscript in its current form fully to assess the work.

In general, the authors have provided a strong rebuttal, including extensive additional analyses and data visualization. While at times one would wish the effects determined had been stronger, or assessed in a more direct manner, I see no procedural or methodological fault in what the authors did. Overall, the impression is that the exposition of the data is true to the nature of the data.

The effort undertaken here in data acquisition and the closed-loop method is, to quote R1, heroic. In the very least (independent of the theoretical impact of the findings) it is a valuable feasibility demonstration that is important to the field.

I have two minor comments that the authors might consider.

1) The authors used structural analysis methods to show that 1) high gamma feature of ECoGs responds to semantic attributes of the movie; 2) the modulation of the inferred vectors by mental imagery is also mainly based on the changes in the high gamma powers. How does this square with studies suggesting that feedback signals (presumably strong in imagery) are predominantly carried by alpha/beta oscillations, whereas it is feedforward signals that dominate in the gamma band?

2) This is a technically demanding paper, and readers might benefit from more guidance throughout the methods section. As the methods are written now the readers might get lost in the details, as the intention of each analysis is not put forward before its description. Adding one or two sentences at the beginning of the paragraphs about the intention of the method described would help to guide the reader. This is a question of taste in scientific writing I invite the authors to consider.

Reviewer #2 (Remarks to the Author):

This is an interesting and ambitious study, which combines intracranial electrocorticography with real-time neuro-feedback and semantic modelling to investigate brain representations of viewed and imagined visual semantic features. A first analysis examined consistency (across repeated movie clips) of response power time-courses in different frequency bands. This identified the high gamma band as most consistent, which became the focus of subsequent analyses. A second set of analyses mapped each scene of a movie into a high-dimensional semantic space, showing that the high-gamma response distinguished scenes related to faces, words and landscapes, and could be used to predict semantic vector direction. A third set of analyses implemented real-time neuro-feedback in a subset of four participants, suggesting that people could drive a stream of images towards semantic vectors representing 'word' and 'landscape' (but less reliably towards 'face'), and that the movement spanned 4-5 dimensions in this space. A final experiment focused on imagery of specific 'word' and 'landscape' images, showing that the neural response moved towards that of the imagined image (and/or away from that of the viewed image) in the second half of each trial when the imagined and perceived categories conflicted. The authors conclude that visual imagery can modulate the neural response to ongoing visual input, and that this depends on the semantic category being imagined.

The study is ambitious and the topic and results are interesting. The methods/analyses are sophisticated and generally appropriate, persuasive, and described in detail. The findings have implications for the understanding of mental imagery as well as for the design of BCI applications. The authors have already thoroughly and carefully addressed a previous round of reviews, adding

useful extra data and many additional analyses and figures. One general comment is that I found some of the results and methods challenging to follow, but this is more to do with their inherent complexity than the descriptions, which seemed generally clear and accurate once I had digested them. Part of the challenge stemmed from relevant information being split between main text, figure legends, methods section and supplementary materials, and needing to match up which dataset/analysis was being described in each case. It might help to ensure that the relevant dataset/analysis is clearly indicated at the beginning of every section, and using consistent terms. (E.g. the "video-watching task" is described in Fig 2. As the "open-loop condition" and it's not always clear which dataset the figures come from. E.g. I think Fig 2d is from the "training video" rather than the "validation video" but this not explicit in the legend.)

I have several more specific comments and queries that the authors could clarify:

- 1) The abstract ends by claiming that the closed loop condition demonstrates an asymmetrical interaction between perception and imagery. While I do not doubt the interaction is asymmetrical, I do not see which aspect of the results demonstrates this.
- 2) For the ANOVAs reported in the text, please add the F statistic and ideally some measure of effect size (e.g. partial eta-squared).
- 3) For the ANOVAs per electrode in Fig 2d, it would be better to correct for multiple comparisons across electrodes (e.g. using false-discovery-rate).
- 4) For the analysis of the ability of semantic/auditory/visual features to predict high-gamma, features (Figure 2e), I assume there will be correlation amongst many of the features (e.g. neighbouring visual gabors). Therefore I would expect autocorrelation amongst the beta coefficients. This means that the effective degrees of freedom when correlating the coefficient vectors is less than the nominal value that would assume independent samples. If so, the autocorrelation would need to be taken into account to get accurate p-values. One possible approach could be Moran Spectral Randomisation (e.g. Wagner & Dray, 2015, <https://besjournals.onlinelibrary.wiley.com/doi/full/10.1111/2041-210X.12407>). (The same issue applies to the correlation between sequences of power estimates in Fig2b, although in this case it may not matter because statistical significance is not being tested.)
- 5) I was curious about the control of the semantic vectors along specific dimensions, as shown in Fig 5g. Can you say anything about what features the significant dimensions might represent? E.g. Mitchell & Cusack, 2016 (<https://www.nature.com/articles/srep20232>), (who also used closed-loop neuro-feedback to study representation of semantic features during perception and imagery, although with fMRI, and without a semantic model) suggested that imagery might drive a subset of the dimensions activated by perception, including more semantic and emotional features than lower-level visual features. That study also suggested that the success of imagery depends on which concept is being imagined, consistent with the present findings of less effective imagery of faces, and delayed imagery of landscapes. This suggests that different feature dimensions might be modulated for imagery of different concepts, in which case averaging across target categories (which I think was done for Fig 5g) could be problematic? Finally, how do you interpret the significant negative correlation of component #6?
- 6) Since the final imagery experiment only uses two categories, it seems that movement of the representation towards the imagined category is confounded with movement away from the perceived category. E.g. words and landscapes lie at the extremes of the semantic space (fig 2c), so if subjects disengaged from the task/stimulus, attended elsewhere, or even closed their eyes, during the imagery phase of each trial, the neural representation might move towards a generic point near the centre of the semantic space, which then happens to be approximately in the direction of the opposite category. In other words, is it possible that the observed effect is driven not so much by active imagery of a specific category, but by reduced attention to the 2nd stimulus?
- 7) A related point is that "word imagery" probably involves a degree of internal verbalisation compared to landscape imagery, so is it possible that the observed effect is driven by verbalisation (or some other process that differs between the two categories) rather than visual imagery per se?
- 8) Page 16 claims that "the accuracy to infer the imagined category was low without feedback, which was reflected in the accuracy at the beginning of the trials ... when the subject imagined images based on the instruction but the feedback screen was black. In contrast, as the feedback continued, the subjects succeeded in controlling the online vector to be closer to the instructed semantic vectors with higher accuracy. Therefore, the data suggested that the closed loop condition improved the accuracy of inferring the imagined category compared to the decoding of

the imagined category in an open-loop condition." However the arrival of feedback is confounded with time since the instruction. I.e. is it possible that accuracy was low at the beginning not because there was not yet feedback, but because it takes some time to construct a reliable mental image?

9) Line 382 states "...during the real-time feedback control, the subjects tended to focus their attention on specific parts of the feedback image..." What is the evidence for this?

10) The acquisition details for the T1 MRI and CT scans are missing. It would also be useful to state when these were acquired relative to electrode implantation.

11) Some figure legends state that "illustrations are shown instead of the actual images used." Is this the case for all figures? If so, this should probably be stated on all figures, or it should be mentioned somewhere that it applies to all figures. It might also be worth stating the reason for this - e.g. copyright restrictions? And perhaps state where the replacement illustrations came from?

12) Figure 4a would be easier to understand if the 4.5 s periods, 250 ms periods, and 1000 ms windows were shown in proportion.

13) In figure 5a, some of the red bars seem wrong. E.g. there are cases when the identical image is repeated but the bar is only placed under one instance.

14) The "DimR" plot at the top-right of Supplementary figure 3a is not explained in the legend.

15) In supplementary figure 4d, why does the correlation converge to a non-zero value?

16) Supplementary figure 7g seems unnecessary.

Point-by-Point Replies for Reviewers

Voluntary control of semantic neural representations by imagery with conflicting visual stimulation

(COMMSBIO-21-2841A)

We appreciate the reviewers' helpful comments and suggestions. We also appreciate the constructive feedback and believe it has helped us clarify and improve the manuscript. We have thoroughly revised our manuscript to address all issues raised by the reviewers. In the revised manuscript, added or corrected text appears in **yellow**. Given the limitation for the length of the manuscript, the current Fig. 5f has been moved to be supplementary Fig. 5e, and part of the discussion was re-written and rearranged. Point-by-point responses to each reviewer's comments (in *gray*) follow.

Reviewer #1 (Remarks to the Author):

Comment 1

I was brought into the review process after the first revision stage. Therefore, I deem it my task to check whether the current response to the previous reviewers is sound, rather than to raise new points. Nevertheless, I have read both the rebuttal and the manuscript in its current form fully to assess the work.

In general, the authors have provided a strong rebuttal, including extensive additional analyses and data visualization. While at times one would wish the effects determined had been stronger, or assessed in a more direct manner, I see no procedural or methodological fault in what the authors did. Overall, the impression is that the exposition of the data is true to the nature of the data.

The effort undertaken here in data acquisition and the closed-loop method is, to quote R1, heroic. In the very least (independent of the theoretical impact of the findings) it is a valuable feasibility demonstration that is important to the field.

Response to comment 1:

We appreciate Reviewer 1's assessment of our study. Responses to the further comments follow.

Comment 2

I have two minor comments that the authors might consider.

1) The authors used structural analysis methods to show that 1) high gamma feature of ECoGs responds to semantic attributes of the movie; 2) the modulation of the inferred vectors by mental imagery is also mainly based on the changes in the high gamma powers. How does this square with studies suggesting that feedback signals (presumably strong in imagery) are predominantly carried by alpha/beta oscillations, whereas it is feedforward signals that dominate in the gamma band?

Response to comment 2:

We appreciate this question. We also feel that the temporal dynamics of the feedforward and feedback signals are important; we therefore compared the decoding accuracy (supplementary Fig. 3e and supplementary Fig. 6e) and the modulation (supplementary Fig. 6f) in multiple frequency bands. Neural representation of the feedforward signals (perception) was observed in all frequency bands, but strongly in the γ band. Moreover, these neural representations were modulated by feedback signals (imagery) for each frequency band, but again strongly in the γ band. Based on these results, we speculate that (1) neural representation for perception and imagery is shared mainly in the γ band, and that (2) sharing occurs in that band because the top-down signals in the α and/or β bands affect the neural representation in the γ band through cross-frequency coupling (Bonnefond & Jensen, 2015; Richter et al., 2017). However, it is true that our results in the other frequency bands also suggest shared neural representation. A study of the temporal dynamics of the feedforward and feedback signals is something we hope to address in future research. Those points are now discussed in the revised manuscript as follows (lines 393–403):

Lastly, the temporal dynamics of the shared neural representation are interesting. Previous studies showed that θ and γ activity convey the bottom-up information, and the α and β activity convey top-down signals^{27, 28}. In line with these studies, our results showed the highest decoding accuracy in the high- γ band (Supplementary Fig. 3e). Moreover, the imagery task revealed that the neural representation in the high- γ band was modulated by the imagery, suggesting shared neural representation between perception and imagery in that band. The suggested explanation is that coupling of the high- γ activity caused by visual stimulus with a lower frequency band such as α ²⁹ or β ³⁰ serves as top-down control for stimulus processing. On the other hand, a recent study using electroencephalograms reported shared neural representations in the α band³¹.

Supplementary Fig .3:

Supplementary Fig. 6:

Comment 3

2) This is a technically demanding paper, and readers might benefit from more guidance throughout the methods section. As the methods are written now the readers might get lost in the details, as the intention of each analysis is not put forward before its description. Adding one or two sentences at the beginning of the paragraphs about the intention of the method described would help to guide the reader. This is a question of taste in scientific writing I invite the authors to consider.

Radoslaw Martin Cichy

Response to comment 3:

We apologize for the lack of descriptions that would guide the reader through the Methods section. Where applicable in the revised manuscript, we added sentences such as the one that follows (lines 814–816) at the beginning of paragraphs to help readers understand the purpose of the analysis about to be described.

To determine the frequency band that responded to the visual stimulus most consistently, consistency of the cortical activity was evaluated using the four repetitions in the validation video.

Reviewer #2 (remarks to the author):

Comment 1

This is an interesting and ambitious study, which combines intracranial electrocorticography with real-time neuro-feedback and semantic modelling to investigate brain representations of viewed and imagined visual semantic features. A first analysis examined consistency (across repeated movie clips) of response power time-courses in different frequency bands. This identified the high gamma band as most consistent, which became the focus of subsequent analyses. A second set of analyses mapped each scene of a movie into a high-dimensional semantic space, showing that the high-gamma response distinguished scenes related to faces, words and landscapes, and could be used to predict semantic vector direction. A third set of analyses implemented real-time neuro-feedback in a subset of four participants, suggesting that people could drive a stream of images towards semantic vectors representing ‘word’ and ‘landscape’ (but less reliably towards ‘face’), and that the movement spanned 4-5 dimensions in this space. A final experiment focused on imagery of specific ‘word’ and ‘landscape’ images, showing that the neural response moved towards that of the imagined image (and/or away from that of the viewed image) in the second half of each trial when the imagined and

perceived categories conflicted. The authors conclude that visual imagery can modulate the neural response to ongoing visual input, and that this depends on the semantic category being imagined.

The study is ambitious and the topic and results are interesting. The methods/analyses are sophisticated and generally appropriate, persuasive, and described in detail. The findings have implications for the understanding of mental imagery as well as for the design of BCI applications. The authors have already thoroughly and carefully addressed a previous round of reviews, adding useful extra data and many additional analyses and figures. One general comment is that I found some of the results and methods challenging to follow, but this is more to do with their inherent complexity than the descriptions, which seemed generally clear and accurate once I had digested them. Part of the challenge stemmed from relevant information being split between main text, figure legends, methods section and supplementary materials, and needing to match up which dataset/analysis was being described in each case. It might help to ensure that the relevant dataset/analysis is clearly indicated at the beginning of every section, and using consistent terms. (E.g. the “video-watching task” is described in Fig 2. As the “open-loop condition” and it’s not always clear which dataset the figures come from. E.g. I think Fig 2d is from the “training video” rather than the “validation video” but this not explicit in the legend.)

Response to comment 1:

We really appreciate Reviewer #2’s interest in the manuscript. First, we apologize for the lack of relevant dataset or analysis information in the figure legends and main text, and the use of the ambiguous term “open-loop”. To increase readability, we added the relevant dataset name for each analysis in the main text and figure legends. Moreover, we removed the confusing term “open-loop” entirely from the manuscript and supplementary information. Our responses to each further point raised by Reviewer #2 follow.

Comment 2

I have several more specific comments and queries that the authors could clarify: 1) The abstract ends by claiming that the closed loop condition demonstrates an asymmetrical interaction between perception and imagery. While I do not doubt the interaction is asymmetrical, I do not see which aspect of the results demonstrates this.

Response to comment 2:

We apologize for the difficulty in the Abstract. What was meant is that modulation in the imagery task was different for word imagery (with landscape perception) and for landscape imagery (with word

perception) during the time from initiation of the imagery. In other words, we observed asymmetrical dependency between categories. To clarify the text, we revised the final two sentences as follows (lines 14–17):

Moreover, modulation of the inferred vectors by mental imagery depended on **the perceived and imagined categories asymmetrically**. The closed-loop control of the semantic vectors revealed **the** asymmetrical interaction between visual perception and imagery.

Comment 3

2) For the ANOVAs reported in the text, please add the F statistic and ideally some measure of effect size (e.g. partial eta-squared).

Response to comment 3:

We appreciate this comment because effect size is also an important measure. In the revised manuscript, F -values for all ANOVAs are reported in the Figures (Fig. 2d, Fig. 5e, and supplementary Fig. 5e [which was Fig. 5f in the earlier version of the manuscript]); colormaps are subsequently used to report each associated partial η^2 in the supplementary figures (supplementary Fig. 2h, supplementary Fig. 5d, and supplementary Fig. 5e respectively). We also added the partial η^2 in the legend of supplementary Fig. 3e, f as follows:

($P < 0.001$, $n = 17$ for each group, $F(4,80) = 8.55$, **partial $\eta^2 = 0.299$** , one-way analysis of variance [ANOVA] with the post hoc Tukey–Kramer test)

($P < 0.001$, $F(18,148) = 8.21$, **partial $\eta^2 = 0.500$** , one-way ANOVA with post hoc Tukey–Kramer tests)

Fig. 2:

Fig. 5:

Supplementary Fig. 2:

Supplementary Fig. 3:

Supplementary Fig. 5:

Comment 4

3) For the ANOVAs per electrode in Fig 2d, it would be better to correct for multiple comparisons across electrodes (e.g. using false-discovery-rate).

Response to comment 4:

We appreciate this comment and agree that correction for multiple comparisons is important. In the revised manuscript, Fig. 2d is updated with p values adjusted using the Benjamini–Hochberg procedure (shown in the response to comment 3).

Comment 5

4) For the analysis of the ability of semantic/auditory/visual features to predict high-gamma, features (Figure 2e), I assume there will be correlation amongst many of the features (e.g. neighbouring visual gabors). Therefore I would expect autocorrelation amongst the beta coefficients. This means that the effective degrees of freedom when correlating the coefficient vectors is less than the nominal value that would assume independent samples. If so, the autocorrelation would need to be taken into account to get accurate p -values. One possible approach could be Moran Spectral Randomisation (e.g. Wagner & Dray, 2015, <https://besjournals.onlinelibrary.wiley.com/doi/full/10.1111/2041-210X.12407>). (The same issue applies to the correlation between sequences of power estimates in Fig2b, although in this case it may not matter because statistical significance is not being tested.)

Response to comment 5:

We agree that autocorrelation is problematic in acquiring correct p values in multiple linear regression analyses. However, instead of multiple linear regression, in those analyses, we used ridge regression with 10-fold nested cross-validation for prediction so as to avoid the problems raised by autocorrelations between features, and we acquired the p values by calculating the correlation coefficient between the true and the predicted brain activity (high- γ features). To clarify our approach, we updated the legend of Fig. 2e as follows (lines 1258–1269):

(e) Pearson's correlation coefficients between the high- γ features while subjects watched the training videos and the inferred high- γ features were color-coded and shown at the location of the electrode. For each outer fold of the 10-fold nested cross-validation, a decoder (ridge regression model) was first trained to infer high- γ features using all features (semantic features, and the low-level visual and auditory features) from the videos; then, for each feature set, the weights

corresponding to the other two feature sets were set to zero before the regression model was applied to the test data. A correlation map of each feature set was calculated from the entire 3,600 scenes. Only electrodes that showed significant positive correlations are shown ($P < 0.05$, $n = 3,600$ for each electrode, one-sided Pearson's correlation test, adjusted using the Benjamini–Hochberg procedure for each feature).

Moreover, as Reviewer #2 pointed out in comment 4, we agree that correction for multiple comparisons is important. Hence, in Fig. 2e, we used the Benjamini–Hochberg procedure to adjust the p values (shown in the response to comment 3; the entire legend is shown above).

Comment 6

5) I was curious about the control of the semantic vectors along specific dimensions, as shown in Fig 5g. Can you say anything about what features the significant dimensions might represent? E.g. Mitchell & Cusack, 2016 (<https://www.nature.com/articles/srep20232>), (who also used closed-loop neuro-feedback to study representation of semantic features during perception and imagery, although with fMRI, and without a semantic model) suggested that imagery might drive a subset of the dimensions activated by perception, including more semantic and emotional features than lower-level visual features. That study also suggested that the success of imagery depends on which concept is being imagined, consistent with the present findings of less effective imagery of faces, and delayed imagery of landscapes. This suggests that different feature dimensions might be modulated for imagery of different concepts, in which case averaging across target categories (which I think was done for Fig 5g) could be problematic? Finally, how do you interpret the significant negative correlation of component #6?

Response to comment 6:

We appreciate this question. In accord with the study by Mitchell & Cusack, 2016, our results also suggest that imagery drives a subset of the dimensions activated by perception (#1, #2, #10, and #15 in Fig. 5f, where there were 14 significant principal components in the video-watching task). Although our data suggest that those components contain information pertaining to lower-level visual, emotional, and semantic features (supplementary Table 2), interpreting the meaning of those components, except for #1 and #2, was difficult. This is a limitation of the study. We now highlight that point in the Discussion (lines 374–381):

In the present study's real-time feedback task, the decoder trained using neural activity during

perception showed significant control for a part of the semantic space (#1, #2, #10, and #15 of 14 significant principal components in the video-watching task). Although interpreting the meaning of the components (except #1 and #2) was difficult (Supplementary Table 2), the results were consistent with the previous study demonstrating that only a subset of the features of the imagined images (e.g., emotional features) are encoded in brain activity when images are imagined than when they are perceived, depending on the imagined category²⁶.

We agree that the success of the imagery depends on the concept being imagined, as Mitchell & Cusack revealed. Actually, our results also suggest that, in the imagery task, modulation for word seems to be larger than that for landscape. During category-based evaluation in the real-time feedback task, modulation for each category has to be evaluated in the same manner. However, in the real-time feedback task, the feedback images were updated every 250 ms, where calculation of the modulation requires 1 s ECoG signals with the subject watching a single image while imagining another image. We were therefore unable to calculate modulation for the real-time feedback task. Nevertheless, we calculated modulation while ignoring three later images during the 1 s time window in which four feedback images were presented and assuming that the subject was watching only 1st image. The result is shown in supplementary Fig. 5c (which appears in the response to comment 3). The result actually suggested a difference in modulation for the various target categories. Averaging across the target categories (word, landscape, and human face) in Fig. 5f (which was Fig. 5g in the earlier version of the manuscript; also shown in the response to comment 3) might therefore lead to underestimation of the modulation. That underestimation is another limitation of the study and is now mentioned in the Discussion (lines 429–436):

The controllability of the inferred image did not seem to depend on the decoding accuracy in the video-watching task but might depend on the semantic attributes. The difference in controllability based on the target category could not be evaluated because of the decoding scheme (Supplementary Fig. 5c); however, our result seems consistent with previous studies suggesting that the high accuracy of identifying perceiving images by perception decoders does not guarantee high accuracy in identifying mental imagery by the decoder^{39,40}.

In the study by Mitchell & Cusack, 2016, they reported negative d' for the dolphin imagery, suggesting that certain features evoke neural representations differently when subjects perceive and imagine the same object (e.g., a dolphin). In our work, we trained the decoder based only on perception. Component #6 therefore seems to be a dimension representing a feature of this type, as now mentioned in the Discussion (lines 381–384):

Interestingly, the previous study also suggested that some of those features activate neural representation differently during perception and imagery, which is in line with our result showing negative correlation during the real-time feedback task (#6 in Fig. 5f).

Comment 7

6) Since the final imagery experiment only uses two categories, it seems that movement of the representation towards the imagined category is confounded with movement away from the perceived category. E.g. words and landscapes lie at the extremes of the semantic space (fig 2c), so if subjects disengaged from the task/stimulus, attended elsewhere, or even closed their eyes, during the imagery phase of each trial, the neural representation might move towards a generic point near the centre of the semantic space, which then happens to be approximately in the direction of the opposite category. In other words, is it possible that the observed effect is driven not so much by active imagery of a specific category, but by reduced attention to the 2nd stimulus?

Response to comment 7:

We appreciate this substantial question. Although we are sure that the subject did not have closed eyes during the task (as monitored by an eye-tracking system or videos recorded during the experiment), we understand the reviewer's concern that the observed effect in the 2nd stimulus might be explained by reduced attention. We therefore calculated the correlation coefficient between the semantic vector of the perceived category and the inferred semantic vector from 0–1 s after image presentation (where evoked responses are within the time window), and we compared the results obtained during the non-imagery period and during the imagery period. The difference between the two periods was close to zero, suggesting that the subject paid the same attention to the 2nd image (in the imagery period) as to the 1st image (in the non-imagery period). That comparison is now shown in supplementary Fig. 6d (as perceived category) and presented in the main manuscript as follows (lines 314–316):

for the difference in the correlation coefficient for the perceived category attributable to the imagery, see Supplementary Fig. 6d

Supplementary Fig. 6:

Comment 8

7) A related point is that “word imagery” probably involves a degree of internal verbalisation compared to landscape imagery, so is it possible that the observed effect is driven by verbalisation (or some other process that differs between the two categories) rather than visual imagery per se?

Response to comment 8:

We appreciate this question. During the real-time feedback task, subjects can use any strategy to control the inferred vector, although we instructed them to visually imagine the instructed category. To control the feedback images, two subjects actually reported that they focused their attention on specific portions of the feedback images that were close to the instructed category (supplementary Table 3). However, because the decoder was trained based on ECoGs from subjects who were simply watching training videos, the strategies that can work during the real-time feedback task are limited to those to which the decoder is sensitive; hence, internal verbalization is unlikely to work in the real-time feedback task (and in the imagery task). We now discuss that point in the revised manuscript (lines 363–368):

Notably, it might be possible to use various strategies other than imagery and attention—for example, internal verbalization—to control the inferred vector. However, because we trained the decoder using the ECoGs while the subjects were simply watching videos, it is unlikely that the trained decoder responded to mental strategies other than vision-related strategies.

Comment 9

8) Page 16 claims that “the accuracy to infer the imagined category was low without feedback, which was reflected in the accuracy at the beginning of the trials ... when the subject imagined images based on the instruction but the feedback screen was black. In contrast, as the feedback continued, the subjects succeeded in controlling the online vector to be closer to the instructed semantic vectors with higher accuracy. Therefore, the data suggested that the closed loop condition improved the accuracy of inferring the imagined category compared to the decoding of the imagined category in an open-loop condition.” However the arrival of feedback is confounded with time since the instruction. I.e. is it possible that accuracy was low at the beginning not because there was not yet feedback, but because it takes some time to construct a reliable mental image?

Response to comment 9:

We understand reviewer 2’s concern that the short interval between the instruction and the arrival of feedback. Although the instruction and the arrival of feedback do not overlap, short interval between them might result in an unreliable mental image and therefore low accuracy for the imagery. To

answer that question, we added classification accuracy for the perceived category during the imagery period of the imagery task (supplementary Fig. 6c; shown in the response to comment 7). The results showed that an imagined category cannot be inferred (AUC in supplementary Fig. 6c subtracted from 1) even in the time window of 1.0–2.0 s after onset of the imagery. Still, modulation of the inferred vector was significant in the same time window (Fig. 7c). In contrast, during the real-time feedback task, the decoders succeeded in inferring the imagined category (41.67%–50.00% in three-choice task), suggesting that closed-loop feedback increases the accuracy with which the imagined category can be inferred. We highlighted that point in the Discussion as follows (lines 449–453):

The low accuracy at the beginning might be also explained by a too short duration to form vivid imagery; however, the decoders also failed to identify the category of the imagined images even in the later time windows of the imagery period in the imagery task (Supplementary Fig. 6c), in which significant modulations were observed.

Comment 10

9) Line 382 states “...during the real-time feedback control, the subjects tended to focus their attention on specific parts of the feedback image...” What is the evidence for this?

Response to comment 10:

We really apologize for the lack of this information. Our evidence comes from subject comments after each session of the real-time feedback task. We now include all of the relevant comments in supplementary Table 3, and a reference to the table has been added to the Results section (lines 238–239) and Discussion section (lines 353–357), respectively, as follows:

for comments from the interviews after each session, see Supplementary Table 3

In fact, during the real-time feedback control, the subjects tended to focus their attention on specific parts of the feedback image that were close to the instructed category (e.g., subtitles in the images when the word instruction was given; see Supplementary Table 3), although we instructed them to imagine the instructed category.

Comment 11

10) The acquisition details for the T1 MRI and CT scans are missing. It would also be useful to state when these were acquired relative to electrode implantation.

Response to comment 11:

We apologize for the lack of information about the MRI and CT scanners. That information has now been added in the revised manuscript. The acquisition time of the images relative to the surgical implantation was also added, as follows (lines 981–994):

Intracranial electrodes were located based on pre-surgical T1-weighted magnetic resonance images (MRIs) and post-surgical computed tomography (CT) images. The MRIs and CT images were acquired at the site where the electrodes had been surgically implanted. The scanners used to obtain the MRIs were the Discovery MR750 (GE Healthcare, Chicago, U.S.A.) for five subjects, the SIGNA Architect (GE Healthcare) for one subject, the Ingenia (Philips Healthcare, Amsterdam, Netherlands) for two subjects, the Achieva (Philips Healthcare) for five subjects, the MAGNETOM Prisma (Siemens, Munich, Germany) for three subjects, the MAGNETOM Skyra (Siemens) for four subjects, and the MAGNETOM Verio (Siemens) for one subject. The CT images were obtained using the Discovery CT750 HD (GE Healthcare) for five subjects, the SOMATOM Emotion 16 (Siemens) for three subjects, the Aquilion ONE (Toshiba Medical Systems, Tochigi, Japan) for four subjects, the Aquilion Precision (Toshiba Medical Systems) for three subjects, and the Aquilion PRIME (Toshiba Medical Systems) for six subjects.

Comment 12

11) Some figure legends state that “illustrations are shown instead of the actual images used.” Is this the case for all figures? If so, this should probably be stated on all figures, or it should be mentioned somewhere that it applies to all figures. It might also be worth stating the reason for this – e.g. copyright restrictions? And perhaps state where the replacement illustrations came from?

Response to comment 12:

We appreciate the comment. Because of copyright restrictions, we arranged for a scientific illustration company to produce original annotated images. An explanation is now included in the legend of the Figure 1, as follows (lines 1238–1241):

Because of copyright restrictions, the actual images used in the tasks have been replaced with illustrations throughout this paper. Details of the creation of the illustrations are presented in the Methods section.

In addition, details about the illustration task are now included in the Methods, as follows (lines 1007–1011):

Illustrations

The graphics of the scenes in this paper were created by an illustration company specialized in scientific illustration and were based on the annotated images from the videos and the annotations, such that the semantic meanings in the annotations were not lost in the resulting illustrations.

Comment 13

12) Figure 4a would be easier to understand if the 4.5 s periods, 250 ms periods, and 1000 ms windows were shown in proportion.

Response to comment 13:

We appreciate this comment. We updated Fig. 4a so that the 250 ms period and the 1,000 ms windows are now proportional. Because the first 2.5 s period in the 4.5 s period is just a blank interval between trials, we shortened it; however, the remainder of the period (2.0 s period in the new figure) is proportional.

Comment 14

13) In figure 5a, some of the red bars seem wrong. E.g. there are cases when the identical image is repeated but the bar is only placed under one instance.

Response to comment 14:

We apologize for the misleading figure, which showed the red bar under the image of the frames in which the three-choice task were considered to be correct. Here, the evaluation of the three-choice task was based on the highest Pearson's correlation coefficient for the online vector against the target vector of the instruction. In contrast, the feedback images were selected based on the highest Pearson's correlation coefficient of the online vector against the semantic vector for each candidate among the 2,926 feedback images. Therefore, even when same feedback image was shown in two frames, the online vector for one frame can be classified as correct (and the image for the frame underlined with red), while the online vector for another frame can be classified as incorrect (and the image for that frame left without an underline). To clarify this point, we added a sentence to the legend for Fig. 5a (shown in the response to comment 3), as follows (lines 1309–1312):

The images underlined in red are the correct decoding in the context of a three-choice task in each frame; because the evaluation of the three-choice task is based on the inferred vector (V_{online}), some frames have the same feedback images, but are differently classified.

Comment 15

14) The “DimR” plot at the top-right of Supplementary figure 3a is not explained in the legend.

Response to comment 15:

We apologize for the lack of an explanation for “*DimR*”. We added the explanation in the legend of supplementary Fig. 3a (shown in response to comment 3):

DimR: dimension-wise correlation coefficients (see Methods).

Comment 16

15) In supplementary figure 4d, why does the correlation converge to a non-zero value?

Response to comment 16:

We appreciate this question raised by reviewer #2. To acquire the inferred vectors in the analysis for supplementary Fig. 4d, we applied a single decoder to the high- γ features obtained during each scene in the validation video (which consisted of four repetition of the 2.5 min movie); then, the consistency of the vectors inferred during the repetitions was evaluated using Pearson’s correlation. Hence, the correlation (consistency) for a certain component can take on a value greater than zero when the decoder predicts similar values for the repeated scenes, even if the decoder could not correctly predict the component. To determine whether that was happening, we shuffled the order of the inferred vectors within each repetition 2,000 times to calculate the chance level of the consistency. The result was that the value for chance-level consistency for all components (including the components that were not significantly decoded) was approximately zero. In the revised manuscript, chance level is now shown in supplementary Fig. 4d, and the permutation method has been given in the legend, as follows:

The gray line indicates chance-level consistency calculated by randomly shuffling the order of the inferred vectors within each repetition of the movies in the validation video 2,000 times.

Supplementary Fig. 4:

Comment 17

16) Supplementary figure 7g seems unnecessary.

Response to comment 17:

We appreciate this comment. Supplementary Fig. 7g is now gone, and the legend for supplementary Fig. 7e, f has been modified as follows:

(e, f) Significant differences in two similarity maps are shown with color coding for higher visual area ($n = 13$) ($P < 0.01$, uncorrected two-sided one-sample t -test). Map (e) shows map (c) – map (b), and map (f) shows map (d) – transverse of map (b). The same analysis performed for the early visual area ($n = 7$) resulted in no significant differences.

Supplementary Fig. 7:

REVIEWERS' COMMENTS:

Reviewer #1 (Remarks to the Author):

The authors have fully addressed all my comments.

Reviewer #2 (Remarks to the Author):

The authors have carefully considered and addressed my comments. My one remaining concern (regarding previous comment #2) is that the final two sentences of the abstract could be still be misinterpreted as referring to asymmetry between perception and imagery, rather than asymmetry between the semantic categories. Otherwise, I congratulate the authors again on an interesting and ambitious study.